# Dual transcriptional activities of PAX3 and PAX7 spatially encode spinal cell fates through distinct gene networks

Robin Rondon[1], Théaud Hezez[1], Julien Richard Albert[1], Shinichiro Hayashi[2,3], Bernadette Drayton-Libotte[2], Gloria Gonzalez Curto[1], Frédéric Auradé[2], Elie Balloul[1], Claire Dugast-Darzacq[1], Frédéric Relaix[2]*, Pascale Gilardi-Hebenstreit[1]*, Vanessa Ribes [1]*

1 Université Paris Cité, CNRS, Institut Jacques Monod, Paris, France, 2 Univ Paris Est Créteil, INSERM, EnVA, EFS, IMRB, Créteil, France, 3 Department of Neuromuscular Research, National Institute of Neuroscience, National Center of Neurology and Psychiatry, Tokyo, Japan

* vanessa.ribes@ijm.fr (VR); pascale.gilardi@ijm.fr (PG-H); frederic.relaix@inserm.fr (FR)

## Abstract

Understanding how transcription factors regulate organized cellular diversity in developing tissues remains a major challenge due to their pleiotropic functions. We addressed this by monitoring and genetically modulating the activity of PAX3 and PAX7 during the specification of neural progenitor pools in the embryonic spinal cord. Using mouse models, we show that the balance between the transcriptional activating and repressing functions of these factors is modulated along the dorsoventral axis and is instructive to the patterning of spinal progenitor pools. By combining loss-of-function experiments with functional genomics in spinal organoids, we demonstrate that PAX-mediated repression and activation rely on distinct *cis*-regulatory genomic modules. This enables both the coexistence of their dual activity in dorsal cell progenitors and the specific control of two major differentiation programs. PAX promote H3K27me3 deposition at silencers to repress ventral identities, while at enhancers, they act as pioneer factors, opening and activating *cis*-regulatory modules to specify dorsal-most identities. Finally, we show that this pioneer activity is restricted to cells exposed to BMP morphogens, ensuring spatial specificity. These findings reveal how PAX proteins, modulated by morphogen gradients, orchestrate neuronal diversity in the spinal cord, providing a robust framework for neural subtype specification.

## Introduction

The generation of an organized array of functionally distinct cell types is a fundamental outcome of development, with profound implications for the physiology of adult organs [1,2]. Transcription factors (TFs), through their ability to activate or repress gene expression, play a pivotal role in patterning cell fates within embryonic tissues

**Data availability statement:** All the raw and processed data associated with this study have been deposited to the NCBI Gene Expression Omnibus (GEO) (GSE288918). pSMAD1/5/9 ChIP-Seq were previously published [doi: 10.7554/elife.104076.2] (GSE275758). Scripts used to process data and generate the figures presented are available under a GNU General Public License v3.0 on GitHub (https://github.com/ribeslab/pax3pax7plosbiol2025) and is available as a Zenodo repository (https://zenodo.org/records/17190900; doi: https://doi.org/10.5281/zenodo.17190807).

**Funding:** PGH is employed by the CNRS; VR, FR and BD are employed by the INSERM. RR received a three-year PhD fellowship from the University of Paris Cité, and his fourth year of PhD was supported by the Labex Who Am I? (ANR-11-LABX-0071). This work was funded by grants to VR from the CNRS/INSERM ATIP-AVENIR program, the Ligue Nationale Contre le Cancer (PREAC2020.LCC/MC; PREAC2016.LCC; RS20/75-114), and the Agence Nationale de la Recherche (ANR) grant AetioSpinoid (ANR-23-CE16-0026-01), as well as by funding to FR from the Labex REVIVE (ANR-10-LABX-73). The funders had no role in study design, data collection and analysis, decision to publish, or preparation of the manuscript.

**Competing interests:** The authors have declared that no competing interests exist.

**Abbreviations :** BMP, bone morphogenetic proteins; BRE, BMP-responsive element; CRM, *cis*-regulatory modules; ESC, embryonic stem cells; NCC, neural crest cells; PWM, position weight matrices; Shh, sonic Hedgehog; TF, transcription factors.

[3,4]. However, for many TFs, the mechanisms by which their pleiotropic activity at *cis*-regulatory modules (CRMs) drive position-specific differentiation within a tissue remain poorly understood [3,4]. This challenge is further compounded by the fact that key TFs exhibit broad expression across cells destined to commit to multiple distinct lineages [3,5].

The developing mammalian spinal cord serves as an ideal model to investigate TF pleiotropic activity due to its remarkable cellular diversity and the significant progress in understanding the role of TFs in regulating this diversity [4]. In this tissue, two opposing morphogen gradients—Sonic Hedgehog (Shh) emanating from the ventral side and bone morphogenetic proteins (BMPs) and Wnts from the dorsal side—provide positional cues that guide the spatially organized expression of TFs in distinct, well-defined bands along the dorsoventral (DV) axis (Fig 1A) [4]. The combined activity of these TFs defines 11 pools of neurogenic progenitors organized along this axis, each of which is committed to differentiation programmes into specific neuronal subtypes (Fig 1A) [4]. This transcriptional regulation ensures the proper formation of locomotor and somatosensory circuits in the adult spinal cord [4]. Among these 11 progenitor pools, six are located in the dorsal part of the spinal cord, designated as dp1 to dp6 in order from the most dorsal to the most ventral, and give rise to dI1 to dI6 interneurons (INs) (Fig 1A) [4]. The remaining five pools are found in the ventral spinal cord, labelled p0, p1, p2, pMN, and p3, and generate V0 to V3 INs and motoneurons (MNs) (Fig 1A) [4].

Among those TFs, we focused on PAX3 and PAX7, two paralogous TFs acting in the dorsal part of the spinal cord. PAX3 is first induced at embryonic day 8.0 (E8) in mice, forming a DV gradient spanning the roof plate, neural crest cells (NCCs), all dorsal progenitors, and, in the brachial region, extending ventrally to the ventral p0 progenitors and occasionally the p1 domain [6–8]. Within 12–24 h, its expression becomes uniform and limited to the dorsal spinal cord (Fig 1A) [6–8]. At this stage, PAX7 is also induced, with lower levels in dp1 and dp2 cells and higher levels in dp3–dp6 cells (Fig 1A) [9]. While the absence of PAX7 does not cause significant spinal cord development defects, the loss of PAX3—and even more so in combination with PAX7—leads to DV patterning defects in the spinal cord [10–13]. The dp4 to dp6 dorsal progenitors give rise to ectopic ventral V0 and/or V1 INs in *Pax3*$^{-/-}$ embryos, with an even greater occurrence in *Pax3*$^{-/-}$; *Pax7*$^{-/-}$ embryos [10]. The fate of dp1 to dp3 progenitors in these mutants remains unexplored. However, the decreased expression of *Neurog2* and *Hes1*—two markers of these progenitors—in the absence of PAX3 suggests a role for PAX in their specification [13,14].

The transcriptional mechanisms by which PAX3 and PAX7 regulate the fate of dorsal progenitor pools remain poorly understood, but evidence highlights their versatility, with PAX proteins capable of both transcriptional activation and repression [7,10,13–15]. By analyzing mutants where PAX3 activity is directed toward either repression or activation, we previously demonstrated that PAX proteins preserve the identities of dp4 to dp6 regions by inhibiting the expression of *Dbx1* and *Prdm12*, key determinants of the ventral p0 and p1 fates [10]. While this repressive activity aligns with evidence that PAX7, like ventral spinal cord specification TFs, can bind in vitro to

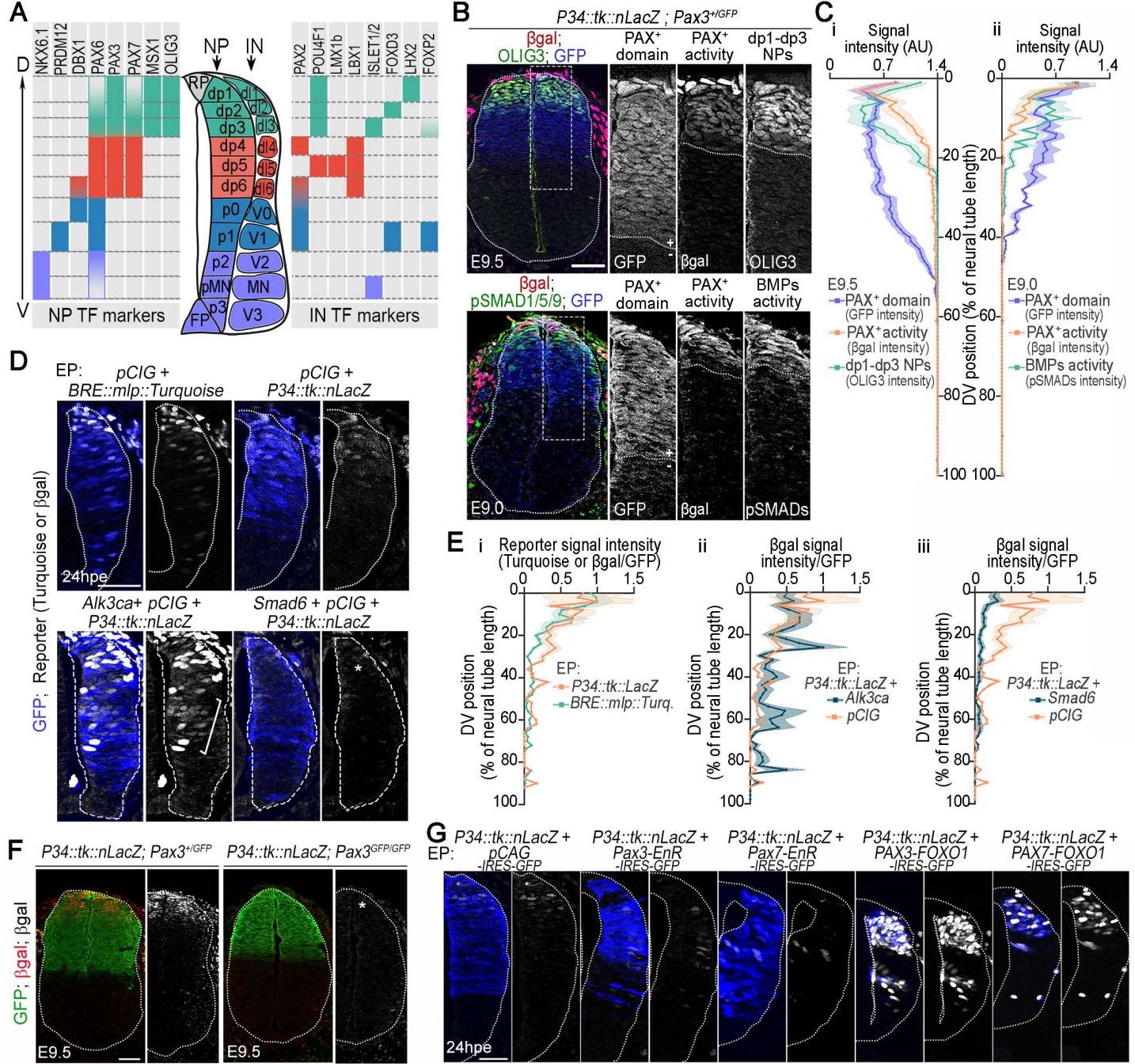

**Fig 1. BMP signaling-dependent dorso-to-ventral gradient of PAX3/7 transcriptional activity revealed by the P34::tk::LacZ reporter. (A)** DV expression patterns of neural progenitor (NP) and interneuron (IN) TF markers in the developing spinal cord. **(B)** Immunostaining for β-galactosidase (β-gal; red/grey), GFP (blue/grey) and either OLIG3 (green/grey) or pSMAD1/5/9 (green/grey) on transverse sections of E9.0 and E9.5 P34::tk::LacZ; Pax3+/GFP spinal cords at brachial level. Black and white panels show magnified views of the boxed region. **(C)** Quantification of the βgal, GFP and OLIG3 **(i)** or pSMAD1/5/9 **(ii)** signal intensity (in arbitrary units, AU) along the DV axis of the neural tube, expressed as the percentage of the neural tube length (bar plots: mean±s.e.m). **(D)** Turquoise fluorescence (white) and immunostaining for GFP (blue) and βgal (white) on transverse sections of chick embryos 24 h post-electroporation (hpe) with the indicated constructs. Only the electroporated side of the neural tube is shown **(E)** Quantification of βgal or Turquoise signal intensity (AU) along the DV axis of the neural tube of chick embryos 24 hpe with the indicated constructs, expressed as a percentage of the neural tube length (mean±s.e.m). **(F)** Immunostaining for β-gal (red/grey) and GFP (green) on transverse sections at the brachial level of E9.5 P34::tk::LacZ; Pax3+/GFP and Pax3GFP/GFP embryos. **(G)** Immunostaining for GFP (blue) and βgal (white) on transverse sections of chick embryos 24 hpe

with the indicated constructs. Only the electroporated side of the neural tube is shown. In all images, white dotted lines outline neural tubes. Scale bars: 50 μm. The data underlying this figure can be found in S7 Table.

co-repressors Groucho/TLE [15], it stands in contrast to PAX3/7 predominant role as transcriptional activators described in other developmental tissues [16]. In these tissues, PAX3 and PAX7 remodel chromatin promoting transcription through diverse mechanisms. PAX7 acts as a pioneer TF capable of binding and opening chromatin regions making them accessible to other TFs [16–18]. It also promotes enhancer-like epigenetic signatures at bound CRMs [19–22], whereas PAX3 facilitates the establishment of three-dimensional chromatin structures, enhancing promoter–enhancer interactions [23]. Such activator functions of PAX are likely to operate in the spinal cord. However, only a few enhancers whose activity depends on PAX3 have been identified near *Pax3*, *Neurog2*, and *Hes1* [7,13,14].

To investigate the transcriptional potential of PAX3 and PAX7 and their role in the DV patterning of spinal cord progenitor pools, we combined genetic and genomic approaches in mouse embryos and embryonic stem cells (ESC) derived organoids. Our findings reveal that the transcriptional activity of PAX3 and PAX7 differs between dp1–dp3 and dp4–dp6 progenitors and depends on the CRMs they target. Across all dorsal progenitors, PAX proteins bind CRMs near ventral identity genes (p0 and p1), acting as repressors and contributing to H3K27me3 deposition. In dp1–dp3 progenitors exposed to BMPs, PAX-mediated repression coexists with transcriptional activation operating at distinct CRMs, promoting chromatin accessibility and driving H3K4me2 deposition and activating gene expression required for dI1–dI3 IN specification. Thus, the dual transcriptional role of PAX3/7, which expands the regulatory capacity of these TFs, constitutes a powerful mechanism for guiding the spatially organized diversification of cell types within the spinal cord.

## Results

### PAX3/7-driven gene activation in the embryonic spinal cord is spatially enabled by BMP signaling

To first investigate the spatial dynamics of PAX3 and PAX7 function as transcriptional activators in the embryonic spinal cord, we used the *P34::tk::nLacZ* reporter transgene [24] (Figs 1B–1G, S1A and S1B). This construct drives *LacZ* expression through a concatemer of five PAX3/7 binding sites recognized by PAX3 and PAX7 paired DNA-binding domain, located upstream of the *thymidine kinase* (*tk*) minimal promoter [24]. Reporter activity was assessed by β-galactosidase immunolabelling, alongside GFP expressed from a *Pax3GFP* knock-in allele, in mouse embryos starting at E9.0—a stage when PAX3 and PAX7 are expressed and progenitors have distinct DV identities until E11.5 (Figs 1B, S1A and S1B). At all stages analyzed, while PAX3-GFP expression spanned the dorsal half of the neural tube, β-galactosidase expression showed a dorsoventral gradient within the PAX3 domain—strong dorsally and absent ventrally (Figs 1B, 1C, S1A and S1B). Co-immunolabelling of β-galactosidase and OLIG3, which marks the dp1–dp3 progenitor domain, showed that the reporter activity's ventral boundary aligned with the dp1–dp3 domain boundary (Figs 1B, 1Ci, S1A and S1B). These results confirm that PAX proteins can function as transcriptional activators in the embryonic spinal cord [7,13,14], but only within dp1–dp3 progenitors—highlighting a context-dependent regulation of their activity.

Since reporter activity was restricted to dp1–dp3 progenitors, we hypothesized that BMP signaling, which specifies these cell types [25–29], might instruct the spatial pattern of PAX3/7 transcriptional activity. To test this, we compared *P34::tk::LacZ* reporter expression with BMP pathway activity, assessed by immunostaining for the phosphorylated forms of SMAD1/5/9, the canonical transcriptional effectors of BMP signaling, at E9.0 (Fig 1B and 1Cii). The spatial profile of reporter activity closely matched the pSMAD1/5/9 gradient. To further assess this spatial alignment, we electroporated the neural tube of chick embryos at Hamburger–Hamilton (HH) stages 10–11 with a BMP-responsive element (BRE) reporter driving Turquoise expression [30], or with the *P34::tk::LacZ* construct. After 24 h post-electroporation (hpe), the spatial patterns of BMP activity and *P34* reporter expression were closely aligned (Fig 1D and 1Ei). To test causality, we manipulated BMP signaling in chick by overexpressing either a constitutively active BMP receptor (ALK3CA) [31] or the pathway

inhibitor SMAD6 [32] together with the GFP-expressing vector *pCAG-ires-GFP* (*pCIG*) to trace electroporated cells (Figs 1D, 1E and S1C). While PAX3 expression remained largely unchanged in both intensity and pattern following ALK3CA or SMAD6 overexpression (S1D Fig), ALK3CA induced scattered ectopic reporter activation in ventral regions (Fig 1D and 1Eii), whereas SMAD6 abolished reporter activity (Fig 1D and 1Eiii). Together, these results demonstrate that BMP signaling promotes PAX-dependent transcriptional activity in specific dorsal regions of the developing neural tube.

We next sought to evaluate the relative contributions of PAX3 and PAX7 to *P34::tk::lacZ* reporter activation (Fig 1F and 1G). We first crossed the *P34::tk::lacZ* mouse line into a *Pax3$^{GFP/GFP}$* background. Reporter expression was not induced in E9.5 *Pax3$^{GFP/GFP}$*, in comparison with *Pax3$^{+/GFP}$*, indicating that PAX3 is required for reporter activation at this stage (Fig 1F). The same analysis could not be performed for PAX7 due to the presence of a *LacZ* transgene in *Pax7* mutant alleles [12]. To circumvent this, we tested the ability of PAX3 and PAX7 variants to activate the *P34::tk::lacZ* reporter in chick neural tubes (Fig 1G). Embryos at HH10–11 were co-electroporated with the *P34::tk::lacZ* reporter and the bicistronic *pCAG-ires-GFP* (*pCIG*) expression vector—either empty or encoding PAX variants, allowing us to trace electroporated cells. These included fusion constructs encoding the PAX3 or PAX7 DNA-binding domain linked to either the FOXO1 transactivation domain (PAX3 or PAX7-FOXO1) or the Engrailed repressor domain (PAX3 or PAX7-EnR) [24,33]. Expression of either PAX3-EnR or PAX7-EnR repressed reporter activity, while both PAX3-FOXO1 and PAX7-FOXO1 strongly activated it, regardless of the dorsoventral position of the electroporated cells (Fig 1G). These findings support a role of both PAX3 and PAX7 in driving *P34::tk::LacZ* reporter activation.

Altogether, these results show that both PAX3 and PAX7 are capable of activating gene expression in the developing neural tube, but that this potential is spatially constrained by BMP signaling. The absence of reporter activity in *Pax3*-null embryos, combined with PAX3's earlier expression and broader dorsal distribution compared to PAX7 [6–9], suggests that PAX3 acts as the primary activator during early dorsal spinal cord patterning. These data highlight a context-dependent mechanism by which BMP signaling permits PAX3/7-dependent transcription specifically within dorsal progenitor domains, thereby linking extracellular positional cues to the emergence of spinal cellular diversity.

## Dorsoventral modulation of PAX3/7 transcriptional activity shapes the emergence and organization of spinal cellular diversity

To assess the role of spatially modulated PAX3/7 activity in IN diversity, we first analyzed the effects of PAX3 and/or PAX7 loss in mouse embryos carrying compound null knock-in alleles for *Pax3* (*Pax3$^{GFP}$*) and/or *Pax7* (*Pax7$^{LacZ}$*) (Figs 2 and S2A) [12,34]. We monitored IN subtypes specification at E11.5 using immunostaining for the TFs LHX2 (dI1; Fig 2A and 2Bi), FOXD3 (dI2, V1; Figs 2A, 2Bii, 2Bvi and S2A), ISLET1/2 (dI3, pMN; Fig 2A and 2Biii), LBX1 (dI4–dI6; Fig 2A and 2Biv), LMX1B (dI5; Fig 2A and 2Bv) and FOXP2 (high in V1; S2A Fig) alongside GFP driven by *Pax3* locus. We focused our analysis on the brachial region of the spinal cord, which remains morphologically preserved in *Pax3* and *Pax3;Pax7* mutant embryos given that neural tube closure defects primarily affect the lumbar region in *Pax3* mutants and extend rostrally into the thoracic region in double mutants [11,35], complicating the assessment of interneuron positioning in those areas.

The number and distribution of INs were the same in *Pax7$^{LacZ/LacZ}$* embryos as in wild-type (WT) or double-heterozygous embryos (*Pax3$^{+/GFP}$; Pax7$^{+/LacZ}$*) (Fig 2A and 2B), consistent with the absence of a central nervous system phenotype in *Pax7* null mutants [12]. In contrast, *Pax3$^{GFP/GFP}$* embryos exhibited a reduced number of dI2 (FOXD3$^+$), dI3 (ISLET1/2$^+$), and dI5 (LMX1B$^+$) INs, reaching between one-third and half the number observed in WT embryos (Fig 2A and 2B). The phenotype of *Pax3$^{GFP/GFP}$; Pax7$^{LacZ/LacZ}$* embryos was more severe, with a reduction of at least two-thirds in the number of all six types of dorsal INs (dI1 to dI6) (Fig 2A and 2B). Moreover, dorsal INs were mispositioned, as exemplified by dI4–dI6 (LBX1$^+$) INs, which were found in the dorsal-most region of the spinal cord and extended into its dorsomedial part, a domain normally occupied exclusively by dI1–dI3 (LBX1$^-$) INs (Fig 2A, arrows). This reduction in dorsal INs was accompanied by an increase in FOXD3$^+$ INs, that were distinguishable by FOXP2$^{high}$, expression [10,11] (Figs 2A, 2Bvi and S2A).

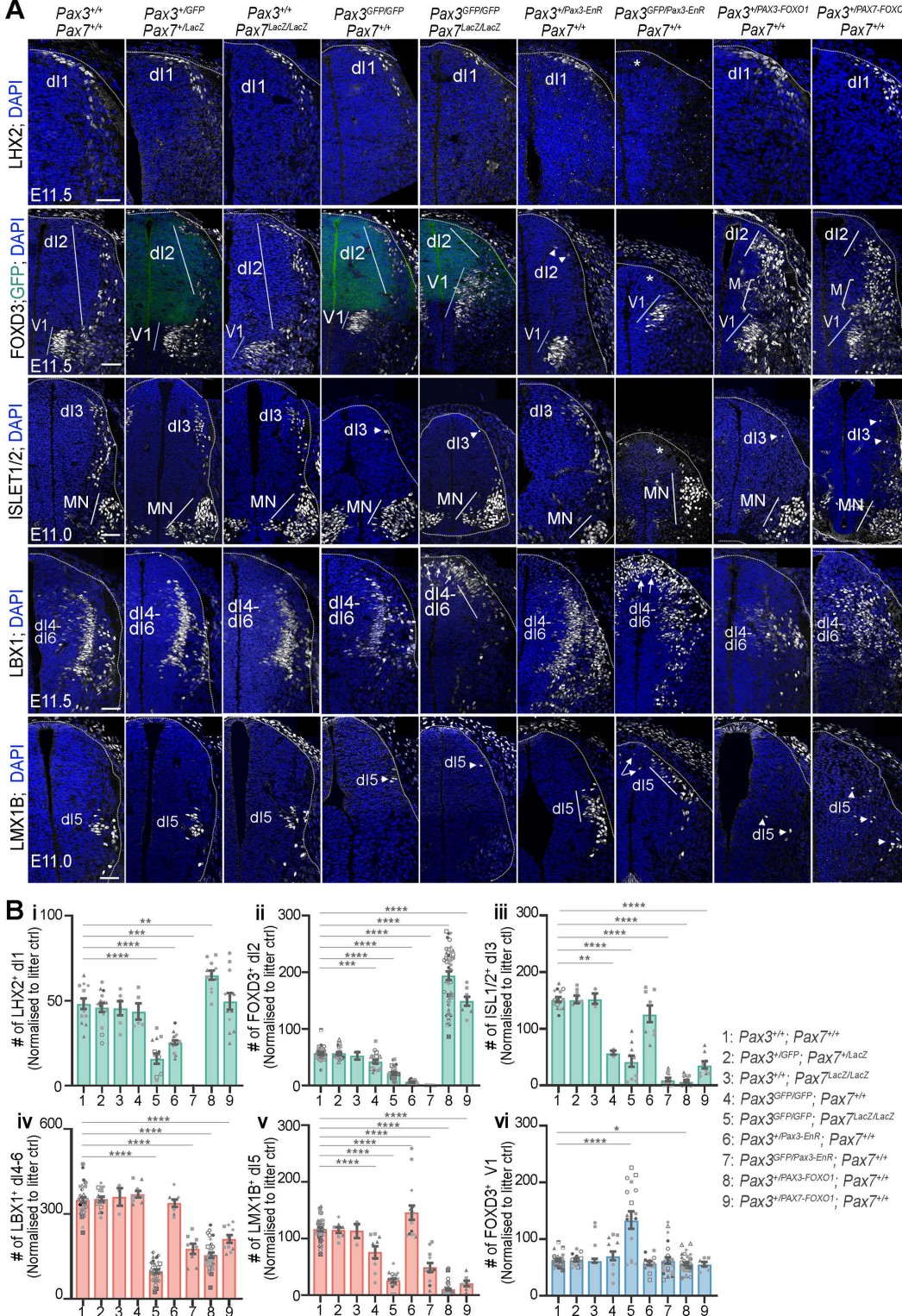

**Fig 2. Neuronal diversity within the spinal cord of mouse embryos following modulation of PAX3 and/or PAX7 transcriptional activity. (A)** Immunostaining for LHX2, FOXD3, ISLET1/2, LBX1 and LMX1B (all in white), GFP (green), and DAPI-stained nuclei (blue) on transverse sections of the brachial region of E11.0–E11.5 mouse embryos with the indicated genotypes. Only half of the neural tube is shown. Asterisks indicate loss of a neuronal

subtype; arrowheads denote a reduced number of cells of a given identity; arrows point to cells with ectopic dorsal position. "M" denotes a mixed population of INs displaying V1 or dI2-like characteristics. White dotted lines outline neural tubes. Scale bars: 50 μm. **(B)** Quantification of cells positive for the indicated TFs (dots: values per transverse section, normalized to those of heterozygous littermates to account for stage variations across littermates, dot shapes: independent embryos; bar plots: mean±s.e.m.; Mann–Whitney U test: *$p < 0.05$; **$p < 0.01$; ***$p < 0.001$; ****$p < 0.0001$). The data underlying this figure can be found in S7 Table.

V1 INs arose not only from ventral progenitors PAX/GFP negative but also from dorsal PAX/GFP expressing progenitors, in a region that normally produces dI4–dI6 (LBX1+) INs (Figs 2A and S2A). This shows a switch of identity for some progenitors from dorsal to ventral. Collectively, these results highlight the partially redundant roles of PAX3 and PAX7 in the production of all dI1–dI6 INs and their spatial distribution along the DV axis of the spinal cord, with PAX3 playing a dominant role in dI1–dI3 IN production.

To determine which of the PAX3/7 transcriptional activities—activator or repressor—contributes to dorsal spinal cord cellular diversity, we analyzed the phenotypic consequences of modifying PAX3 transcriptional potential. Specifically, we examined IN populations in E11.5 mouse embryos carrying conditional *Pax3* knock-in alleles expressing either PAX3-FOXO1 or PAX3-EnR, in combination with a *Pax3+* or a *Pax3GFP* allele (Fig 2A and 2B) [24,33]. In addition, we analyzed embryos carrying a newly generated conditional *Pax3* knock-in allele expressing PAX7-FOXO1 fusion protein (Fig 2A and 2B) (see "Materials and methods", S2B Fig). This fusion protein consists of the N-terminal portion of human PAX7— including the paired and homeodomain DNA-binding domains—fused to the same C-terminal FOXO1 moiety used in the PAX3-FOXO1 construct, this corresponds to the PAX7-FOXO1 fusion protein associated with certain forms of alveolar rhabdomyosarcoma [36]. We confirmed expression of the PAX7-FOXO1 fusion protein in *Pax3+/PAX7-FOXO1* embryos by immunostaining for the C-terminal FOXO1 moiety, which marks cells of the *Pax3* lineage (S2C Fig). This new mouse line enabled us to assess whether PAX7-associated transcriptional activity induces distinct cell fate outcomes compared to PAX3, when expressed from the same genomic locus and thus under equivalent expression levels and within the same cellular context. As with the loss-of-function mutants, all analyses were performed at brachial levels, where the neural tube remains closed [24,33,37].

Strikingly, expressing the repressor PAX3-EnR in *Pax3+* dorsal spinal progenitors permitted differentiation into dI4– dI6 INs while inhibiting dI1 to dI3 identities. In *Pax3+/Pax3-EnR* E11.5 embryos, dI1 INs (LHX2+) and dI2 (FOXD3+) INs were reduced (Fig 2A, 2Bi and 2Bii), whereas dI3–dI6 (ISLET1/2+, LBX1+, LMX1B+) populations remained unchanged (Fig 2A and 2Biii–2Bv). Notably, in *Pax3GFP/Pax3-EnR* embryos, all dI1–dI3 INs were absent (Fig 2A and 2B), and dI4–dI6 INs were generated, they were fewer and abnormally localized dorsally (Fig 2A and 2B). Conversely, the PAX3–FOXO1 activator reduced the generation of dI4–dI6 interneurons, as indicated by fewer dI4–dI6 (LBX1+) INs and dI5 (LMX1B+) INs in E11.5 *PAX3+/PAX3-FOXO1* embryos compared with wild-type (WT) embryos (Fig 2A, 2Biv and 2Bv). Additionally, in the dorsal-most region, dI3 (ISLET1/2+) IN production was reduced (Fig 2A and 2Biii), while dI1 (LHX2+) and dI2 (FOXD3+) INs were more abundant and extended ventrally compared to controls (Figs 2A, 2Bi, 2Bii and S2A). Moreover, ectopic FOXD3+ neurons emerged from the ventricular (progenitor) region, from where LBX1+ dI4–dI6 INs typically arise. Among these, some expressed high levels of FOXP2, marking them as V1 INs, while others, lacking or expressing low FOXP2, resembled dI2-like cells (S2A Fig). Expression of the PAX7-FOXO1 activator induced phenotypes broadly similar to those observed with PAX3-FOXO1 expression, though generally less pronounced (Fig 2A and 2Bi–2Bvi). For example, LHX2+ dI1 INs were not significantly increased in E11.5 *Pax3+/PAX7-FOXO1* embryos, although some embryos exhibited elevated LHX2+ cell numbers compared to controls (Fig 2A and 2Bi). In contrast, the ectopic generation of FOXD3+ dI2 cells (Fig 2A and 2Bii), and the reduction of LBX1+ dI4/dI5 and LMX1B+ dI5 populations (Fig 2A, 2Biv and 2Bv), indicate that PAX7-FOXO1 retains functional activity similar to that of PAX3-FOXO1.

In summary, a forced PAX repressive activity impairs dI1–dI3 IN production, while a forced activation partially skews dp4–dp6 progenitors toward a V1-like or dI2-like fate. Hence, the balance between the activator and repressor activities

of PAX3 and PAX7, regulated along the DV axis, is crucial for shaping the generation of dorsal interneuron subtypes. This underscores the importance of identifying the *cis*-regulatory elements through which these TFs fine-tune distinct transcriptional programs along the DV axis of the spinal cord.

## mESC-derived organoid models unveil spinal IN specification through dual PAX3/7 transcriptional dynamics

To dissect PAX3/7 dual activity in dorsal progenitors, we generated mESC-derived spinal organoids (Fig 3A), overcoming the challenge of accessing limited progenitor pools in embryos while recapitulating their differentiation dynamics [27]. Spinal organoids transitioned from pluripotency (*Klf4+; Pou5f1+; Fgf5low*) to an epiblast state (*Klf4-; Pou5f1+; Fgf5High*) within 48 h, then to a caudal neural state (*Nkx1.2+; Cdx2+; Sox1low; Pax6low*) by 66h (S3A Fig). By day 3, *Sox1* and *Pax6* increased as *Nkx1.2* and *Cdx2* decreased, with *Tubb3+* neurons emerging by day 6 and peaking at day 7 (S3A Fig). DV identities were modulated by BMP4 exposure between days 3–4 (Figs 3A, 3B, S3A and S3B). Untreated organoids (∅) contained mixed dp4–dp6 and ventral p0/p1 progenitors (*Pax3/7+; Olig3-* dp4–dp6*; Prdm12+* p1; *Dbx1+* p0, dp6; *Lbx1+* dl4–dl6) (Figs 3B, S3A and S3B), while BMP4 favored dp1–dp3 fates progenitors (*Pax3/7+; Olig3+* dp1–dp3; *Pou4f1+* dl1–dl3, dl5) (Figs 3B, S3A and S3B). PAX3 and PAX7 were detected from day 3, but their expression varied: BMP4 increased PAX3 while slightly reducing PAX7 (Figs 3B–3D, S3A and S3B). Analysis of *P34::tk::nLacZ* transgene activity in spinal organoids confirmed that PAX3/7 transcriptional activation occurs in this model system only in the presence of BMP4 (Fig 3E). As in vivo, activity was primarily restricted to OLIG3+ progenitors, with approximately 80% of OLIG3+ cells showing β-galactosidase expression (Fig 3E). A small subset of β-galhigh/OLIG3- cells (approximately 5%), likely corresponding to neural crest cells, was also detected in the organoids (arrowheads in Fig 3E). Thus, in spinal organoids, PAX3/7 transcriptional activity is largely confined to dp1 and dp3 progenitors, consistent with its spatial restriction in developing embryos (Fig 1B).

Using CRISPR, we generated three independent *Pax3−/−*, *Pax7−/−*, and *Pax3; Pax7* double knockout (DKO) mESC lines, by excising the first two exons and part of the third exon of the *Pax3* and/or *Pax7* genes. Immunostaining confirmed the absence of both proteins in day 5 mutant organoids (S3C Fig). In all mutant organoids, neural commitment remained intact, with SOX1+ cells exceeding 96% at day5 (S3E Fig). In the absence of one paralog, the other was still expressed. Given that these paralogs are restricted to dorsal progenitors, this demonstrates that progenitor fate remained predominantly dorsal (S3D Fig). The unreduced number of HuC/D+ neurons at day 7 indicating that terminal differentiation remained unaffected in absence of the PAX (S3F Fig).

The cellular profile of *Pax7−/−* organoids treated with BMP4 closely resembled that of controls, with a predominance of LHX2+dl1 INs and very few LBX1+dl4–dl6 INs (Fig 3F and 3Gi–3Gii). However, some differences were noted: the percentage of LHX2+ dl1 interneurons exceeded 30% in *Pax7−/−* organoids, compared to typically less than 30% in controls (Fig 3F and 3Gi). Conversely, the proportion of LBX1+ dl4–dl6 cells remained very low, generally below the 7% seen in controls (Fig 3F and 3Gii). In contrast, *Pax3−/−* organoids exhibited a 6-fold reduction in LHX2+ dl1 INs (Fig 3F and 3Gi), while LBX1+ dl4–dl6 INs increased, comprising nearly 75% of HuC/D+ neurons versus <10% in controls (Fig 3F and 3Gii). The combined loss of PAX3 and PAX7 (DKO) further exacerbated the ventralisation of cell identities. LHX2 expression was nearly absent (Fig 3F and 3Gi), while LBX1+ dl4–dl6 INs made up half of HuC/D+ neurons (Fig 3F and 3Gii). Additionally, 30% of neurons were FOXD3+ (Fig 3F, 3Giii and S4A), but lacked POU4F1, indicating a V1-like rather than dl2-like IN identity (S4A Fig). The presence of PAX2+; LBX1+ V0 and V1-like INs in DKO, absent in WT organoids further confirmed this ventral identity shift (S4B Fig). To determine whether the ventralisation observed in BMP4-treated *Pax3−/−* and DKO organoids resulted from impaired BMP4 response—essential for dl1–dl3 specification [25–29]—we analyzed phosphorylated SMAD1/5/9. One hour post-BMP exposure, when signaling peaks [27], phosphorylated SMAD1/5/9 levels at the organoid periphery were comparable across all genotypes (S4C Fig), ruling out BMP signaling defects as the cause of ventralisation in *Pax3−/−* and DKO organoids.

Even without BMP4, loss of PAX3 or PAX7 loss, and especially their combined loss, led to ventralisation of neuronal identities (Figs 3F, 3G, S4A and S4B). LBX1+ dl4–dl6 INs decreased from 50% in controls to 30% in single mutants and

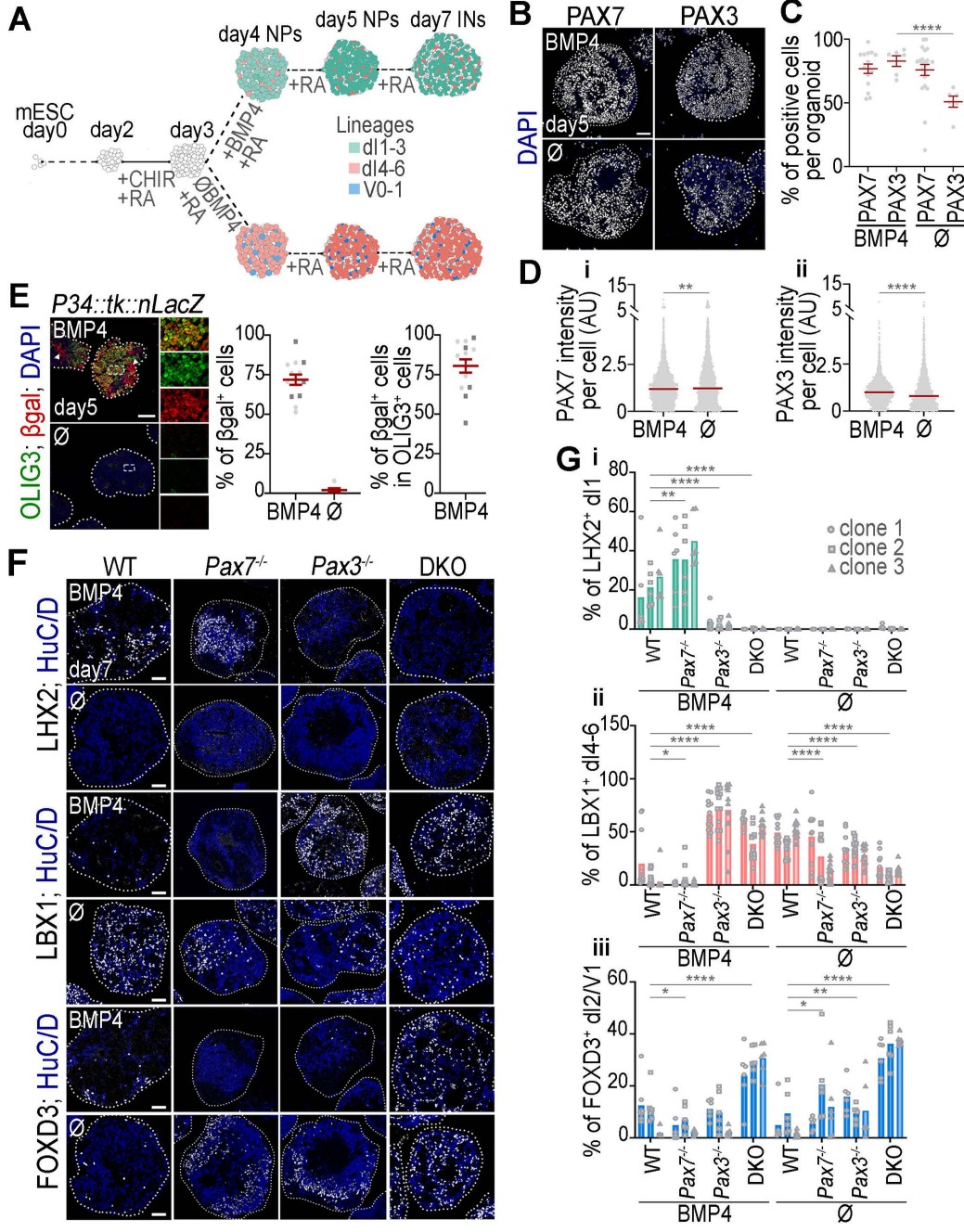

**Fig 3. Expression, activity, and function of PAX3 and PAX7 during spinal neuronal diversity generation in organoids treated or not with BMP4.**
**(A)** Schematic of the differentiation protocol used to generate spinal organoids from mESCs. BMP4-free organoids (∅ condition) are enriched in dp4–dp6 progenitors by day 5 and their associated interneurons (INs) by day 7. BMP4 treatment from day 3 to day 4 shifts differentiation toward dp1–dp3 progenitors and corresponding INs. Only a few ventral V0 and V1 lineage cells are present in untreated organoids. **(B)** Immunostaining for PAX7 or PAX3 (white) and DAPI-stained nuclei (blue) on sections of day 5 WT spinal organoids, treated or not with BMP4. **(C)** Quantification of the percentage of PAX7+ or PAX3+ cells per organoid (dots: individual organoid values; bar plots: mean ± s.e.m.) **(D)** Quantification of PAX7 (**i**) and PAX3 (**ii**) nuclear intensity levels (dots: individual nuclear values in arbitrary units (AU); bar plots: mean ± s.e.m.; Mann–Whitney U test: **p < 0.01; ****p < 0.0001). **(E)** **Images:** Immunostaining for β-galactosidase (βgal; red), OLIG3 (green) and DAPI-stained nuclei (blue) on sections of day 5 *P34::tk::LacZ* spinal organoids treated or not with BMP4 (arrows points to neural crest cells). Right panels are magnified views of the squares shown in the left panels. **Graphs:** Quantification of the percentage of βgal+ cells per day 5 organoids treated or not with BMP4 (left) (dots: organoids and bar plots: mean ± s.e.m.) and

of the percentage of βgal$^+$ cells within the OLIG3$^+$ population in organoid treated with BMP4 (right) (dots: organoids and bar plots: mean±s.e.m.). **(F)** Immunostaining for LHX2, LBX1, and FOXD3 (all in white) and DAPI stained nuclei (blue) on sections of day 7 spinal organoids with the indicated genotypes, treated or not BMP4. **(G)** Quantification of the percentage of cells expressing LHX2 (i), LBX1 (ii), or FOXD3 (iii) in day 7 organoids of the indicated genotypes (3 independent clones per genotype). (dots: values per transverse section, dots shapes: independent clones; bar plots: mean±s.e.m.; Two-way ANOVA: *$p<0.05$; **$p<0.01$; ***$p<0.001$; ****$p<0.0001$). WT: Wild-type; DKO: *Pax3; Pax7* double mutant. In all images, white dotted lines outline organoids. Scale bars: 50 μm. The data underlying this figure can be found in S7 Table.

below 10% in DKO organoids (Fig 3F and 3Gii). Conversely, V0 and V1 INs (FOXD3$^+$, PAX2$^+$, POU4F1$^-$) increased from approximately 10% in controls to over 15% in single mutants and up to 30% in DKO organoids (Figs 3F, 3Giii, S4A and S4B).

In conclusion, as in embryos, PAX3/7 transcriptional activation in organoids is restricted to dp1–dp3 progenitors and absent in dp4–dp6. In both models, PAX3 and PAX7 cooperate to prevent ventral identity acquisition, with PAX3 playing a dominant role in dp1–dp3. However, unlike in vivo, PAX7 loss significantly impacts dI4–dI6 IN production in organoids. This discrepancy may stem from reduced PAX3 expression in dp4–dp6 cells in vitro, likely due to the need for finer BMP and Wnt signaling modulation during differentiation [7,31,38].

### *PAX*3 and PAX7 differentially shape dorsoventral gene regulatory networks

To investigate PAX3/7-regulated transcriptional programs in dp1–dp3 and dp4–dp6 progenitors, RNA-seq was performed on WT, *Pax3$^{-/-}$*, *Pax7$^{-/-}$*, and DKO organoids (Fig 4A–4F). Using three independent cell lines per genotype, organoids were treated or untreated with BMP4 and harvested at day 6, when progenitors and INs coexist (Fig 4A and S1 Table).

Principal component analysis revealed that transcriptomic profiles were primarily driven by BMP4 treatment, with genotype-specific variation emerging clearly only in BMP4-treated organoids (Fig 4A). Accordingly, DESeq2 analysis identified a substantially larger number of differentially expressed genes (DEGs) between control and *Pax3*/*Pax7* mutants in BMP4-treated conditions than in untreated conditions (Fig 4B and S1 Table). These DEGs included both PAX-activated (downregulated in DKO) and PAX-repressed (upregulated in DKO) genes (Fig 4B). This pattern reflects a stronger regulatory role for PAX3/7 in dp1–dp3 progenitors (induced by BMP4) compared to dp4–dp6 progenitors (Ø condition). Notably, the number of DEGs was highest in DKO organoids, exceeding that in either single mutant, suggesting partial redundancy between PAX3 and PAX7 (Fig 4C and 4D). Under BMP4 conditions, genes deregulated in DKO organoids were partially affected in *Pax3$^{-/-}$*, but not in *Pax7$^{-/-}$* organoids, confirming that PAX3 is the primary transcriptional regulator in this context (Fig 4C). In the absence of BMP4, gene expression in single mutants was intermediate between WT and DKO, suggesting that PAX3 and PAX7 regulate overlapping gene networks (Fig 4D).

To further characterize PAX3/7 target programmes, we defined gene signatures for dp1–dp3, dp4–dp6, p0–p1 progenitors, along with their associated interneurons (INs), using single-cell RNA-seq and histological data from embryonic mouse and chicken spinal cords [4,39,40]. We then examined the enrichment for these molecular signatures among DEGs upon *Pax3* and *Pax7* loss (Fig 4E and 4F and S1 Table). In BMP4-treated organoids, PAX3/7-activated genes were highly enriched in dp1–dp3/dI1–dI3 signatures (Fig 4E). These included key transcription factors such as *Olig3* (required for dI2/dI3 INs), *Atoh1*, and *Msx1* (critical for dI1 INs), as well as postmitotic markers *Lhx2*, *Pou4f1*, and *Isl1* [4,41–43] (Fig 4F and S1 Table). To validate these transcriptomic effects, we assessed OLIG3 protein expression in day 5 organoids and E9.5 mouse embryos (Fig 4G and 4H). In BMP4-treated WT organoids, OLIG3$^+$ cells comprised more than 60% of all cells, whereas OLIG3 was absent in untreated organoids (Fig 4G). In contrast, OLIG3 induction was abolished in double knockout (DKO) organoids even in the presence of BMP4 (Fig 4G), consistent with RNA-seq data. A comparable reduction was observed in E9.5 *Pax3; Pax7* double mutant embryos where OLIG3$^+$ cell numbers decreased 6-fold compared to controls (Fig 4H). Interestingly, OLIG3 expression was diminished but not eliminated in *Pax3$^{-/-}$* organoids (Fig 4G), indicating that PAX3 is the primary regulator of OLIG3, with PAX7 contributing when PAX3 is absent (Fig 4G).

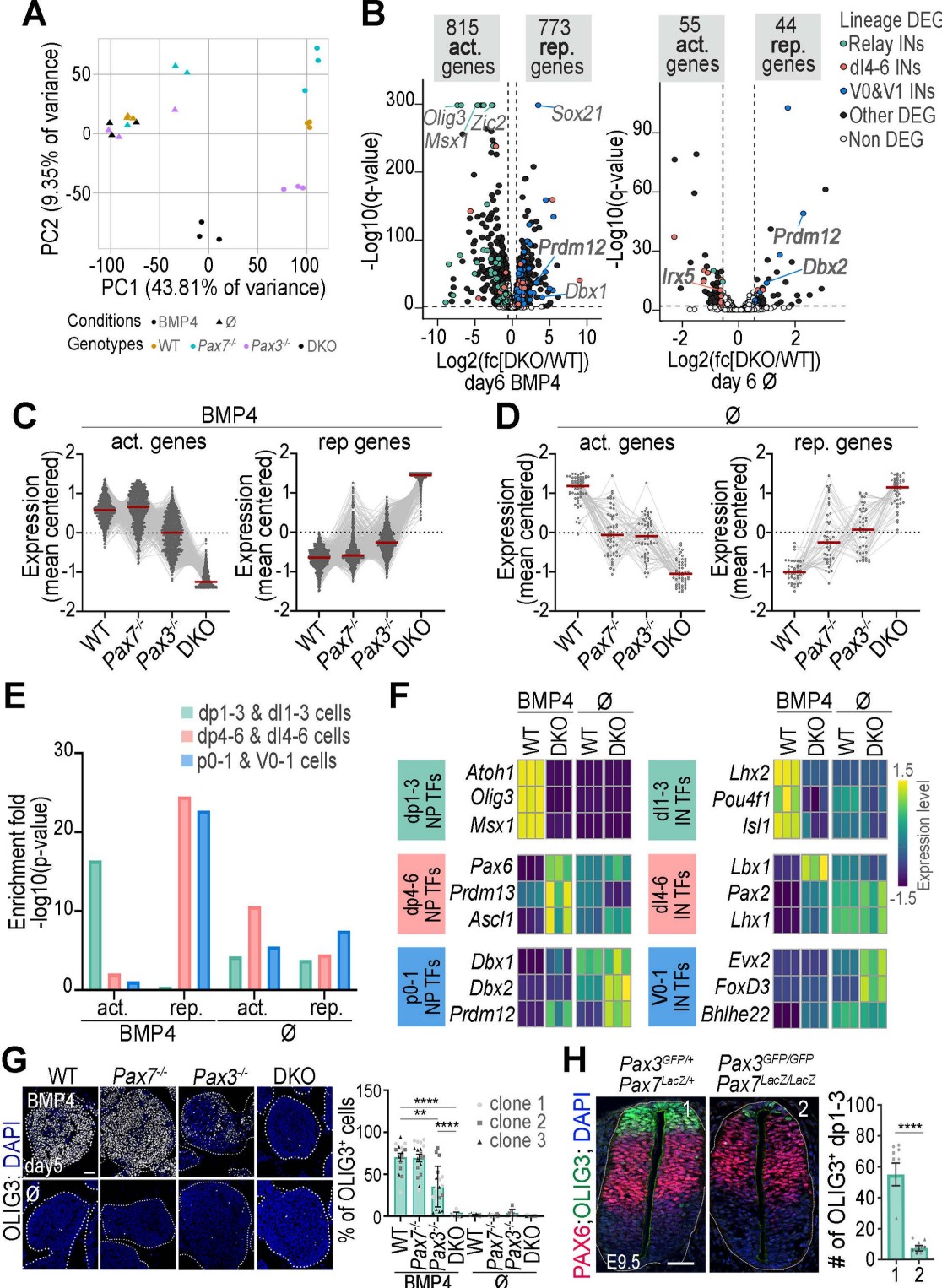

**Fig 4. Transcriptional programmes regulated by PAX3 and PAX7 activity in organoids treated or not to BMP4.** **(A)** Principal component analysis of day 6 organoids with the specified genotype, treated or not with BMP4. **(B)** Volcano plots comparing gene expression between WT and DKO day 6 organoids, treated or not with BMP4. The *y*-axis shows statistical significance ($-\log_{10}(q$-value)) and the *x*-axis represents $\log_2$(fold change). White dots:

non-significant differentially expressed genes (DEG), black/colored dots: significant DEGs. Blue, pink, and green dots highlight master TFs for ventral, dI1–dI3, and dI4–dI6 IN lineages, respectively. **(C–D)** Normalized mean-centered expression levels of genes significantly downregulated or upregulated in DKO versus WT day6 organoids, treated **(C)** or not **(D)** with BMP4, shown for WT, *Pax7*$^{-/-}$, *Pax3*$^{-/-}$, and DKO organoids (dots/lines: individual genes; bars: mean±s.e.m.; $n=3$ clones/genotype). **(E)** Enrichment of genes encoding specifiers of dp1–dp3, dp4–dp6 and p0–p1 progenitors (NPs) and their corresponding INs, among genes downregulated (act.) or upregulated (rep.) in DKO organoids compared to WT day 6 organoids, with or without BMP4. **(F)** Heatmaps showing normalized, mean-centered mRNA expression of TFs marking dp1–dp3, dp4–dp6 and p0–p1 NPs and their associated INs assessed by RNA-seq in day 6 WT or DKO organoids, treated or not with BMP4, in 3 independent experiments. Fold change across samples are color-coded in blue (lower levels) to yellow (higher levels). **(G)** Immunostaining for OLIG3 (white) and DAPI-stained nuclei (blue) on sections of day 5 spinal organoids with the indicated genotype, treated or not with BMP4. White dotted lines outline organoids. Scale bar: 50 µm. **Graph:** Quantification of the percentage of OLIG3$^+$ cells in these organoids (dots: values per transverse section, dots shapes: independent clones; bar plots: mean±s.e.m; Mann–Whitney U test: ***$p<0.001$; ****$p<0.0001$; ns: non-significant). **(H)** Immunostaining for OLIG3 (green), PAX6 (red) and DAPI stained nuclei (blue) on transverse sections of E9.5 mouse embryos with the indicated genotypes. White dotted lines outline neural tubes. Scale bar: 50 µm. **Graph:** quantification of the number of OLIG3$^+$ cells in such embryos (dots: values per transverse section, dots shapes: independent embryos; bar plots: mean±s.e.m; Mann–Whitney U test: ****: $p<0.0001$). The data underlying this figure can be found in S1 and S7 Tables.

Conversely, PAX3/7-repressed genes in BMP4-treated organoids were enriched in dp4–dp6/dI4–dI6 and p0–p1/V0–V1 signatures (Fig 4E). These included *Pax6* (expressed in intermediate spinal cord regions [44], *Prdm13* and *Ascl1* (which drive dI4–dI6 INs differentiation [45,46]), and IN markers *Lbx1*, *Pax2*, and *Lhx1* (dI4–dI6) [4] (Fig 4F). In addition, p0/p1 regulators *Prdm12*, *Dbx1, Dbx2* [47,48] and V0/V1 IN markers *Foxd3, bHlhe22*, and *Evx2* were also de-repressed in the absence of PAX factors. Similar, though less pronounced, enrichments were observed in untreated organoids, with PAX-repressed genes predominantly associated with p0–p1 and V0/V1 lineages, and PAX-activated genes enriched in dp4–dp6 signatures (Fig 4E and 4F).

These findings demonstrate that PAX3 and PAX7 orchestrate largely overlapping transcriptional programs essential for the specification of dp1–dp3, dp4–dp6, and p0–p1 progenitors, with PAX3 playing a more prominent role, particularly in dp1–dp3 fate specification. PAX factors universally repress p0–p1 programs, preventing dorsal progenitors from acquiring V0/V1 identities across conditions. In contrast, dp1–dp3 programs, exclusive to BMP4 conditions, rely entirely on PAX3/7 activation. Finally, PAX3/7 promote dp4–dp6 programmes in absence of BMP4 but repress them under BMP4 stimulation, reflecting BMP-dependent modulation of PAX-regulated networks.

## PAX3/7 drive distinct chromatin state dynamics along the DV axis

To uncover CRMs driving PAX3/7 transcriptional effects, we generated mESC lines with 3xFLAG-tagged *Pax3* or *Pax7*, enabling CUT&Tag-based profiling of genomic recruitment sites in day 5 organoids (Fig 5A and S2 Table). We identified 3,297 PAX3/7-enriched peaks, primarily in intronic and distal intergenic regions (Figs 5A and S5A and S2 Table). Using GREAT [49], we mapped nearby genes and found significant enrichment among both PAX3/7-activated and repressed transcriptomic targets, suggesting these regions act as CRMs, promoting or inhibiting gene expression (S5B Fig).

PAX3 and PAX7 exhibited broadly similar recruitment patterns, with differences in peak intensities reflecting their expression: PAX3 peaks were stronger in BMP4-treated organoids, while PAX7 peaks stronger in untreated ones (Fig 5A). Notably, recruitment sites differed significantly with BMP4 treatment, highlighting context-dependent binding (Fig 5A and 5B). Of the 3,297 PAX3/7-bound CRMs, 2,835 were shared across conditions, for example two within *Prdm12* locus (p1 marker) (Fig 5A and 5B). BMP4-treated organoids had 282 unique CRMs, such as near *Msx1* (dp1–dp3), whereas 180 CRMs were exclusive to untreated organoids, for instance intronic regions of *Prdm8* (ventral marker)(Fig 5A and 5B).

To understand whether DNA sequence features contribute to context-specific recruitment, we performed motif enrichment analysis across three CRM groups: BMP4-specific, shared, and untreated-specific (S5C, S5D and S5H Fig). First, we examined motifs recognized by PAX DNA-binding domains—the Paired domain and Homeodomain [50]. These included motifs bound by the Paired domain (PrD), a Homeodomain dimer (HD–HD), or both domains simultaneously (PrD–HD) (S5C Fig) [16]. Shared CRMs were strongly enriched for all PAX motifs, particularly the composite PrD–HD motif (S5D Fig).

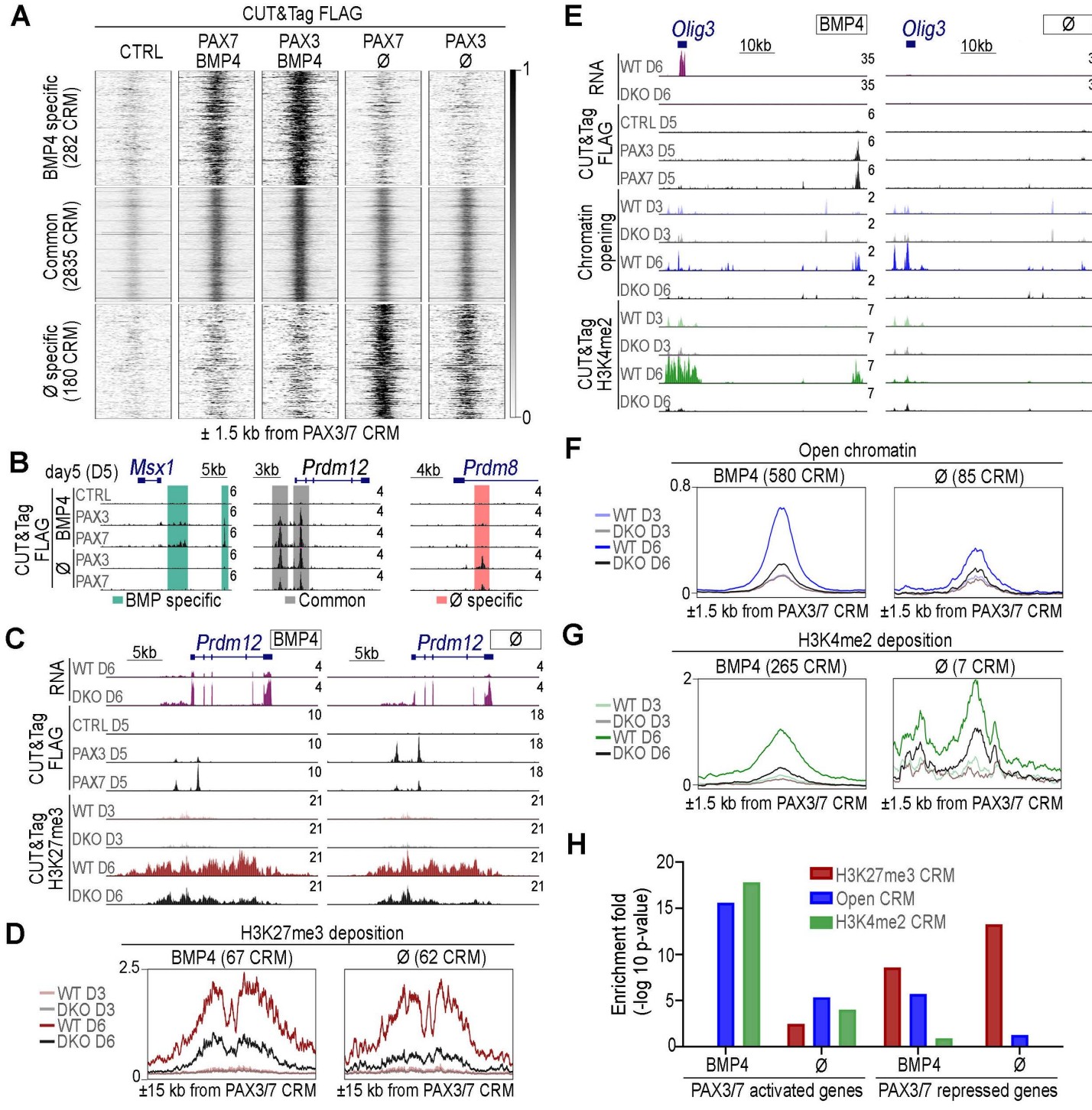

**Fig 5. Dynamics of chromatin states at PAX3/7-bound CRMs following PAX3/7 loss and BMP4 exposure. (A)** Heatmaps of normalized FLAG CUT&Tag signals for all PAX3/7-bound CRMs identified in organoids expressing FLAG-PAX3 (PAX3), FLAG-PAX7 (PAX7), or not (CTRL), treated or untreated with BMP4 at day 5 of differentiation. Signals are shown over a 3 kb window centered on CRM midpoints (*n* = 3,297). Three CRM groups are highlighted: those specific to BMP4-treated organoids, specific to untreated organoids, and shared between both conditions. **(B)** UCSC Genome Browser screenshots showing normalised FLAG CUT&Tag read distributions in wild-type (CTRL) and FLAG-PAX3 (PAX3) or FLAG-PAX7 (PAX7) organoids, with or without BMP4 treatment, at day 5 of differentiation. Signal is shown in counts per million (CPM). Coordinates: chr5:37816721-37837786

(left, scale bar: 5 kb), chr2:31628811-31658766 (middle, scale bar: 3 kb), and chr5:98159967-98180223 (right, scale bar: 4 kb). Shared binding sites are highlighted in grey, BMP4-specific in green, and untreated-specific in salmon pink. **(C)** UCSC Genome Browser screenshots showing normalised RNA-seq (RNA), FLAG, or H3K27me3 CUT&Tag read distributions in wild-type (CTRL), FLAG-PAX3 (PAX3) or FLAG-PAX7 (PAX7) expressing, and *Pax3; Pax7* double-knockout (DKO) organoids, at days 3, 5 or 6 (D3, D5, D6) of differentiation, treated or not BMP4. Signal is shown in CPM. Coordinates: chr2:31627589-31,663,216, scale bar: 5 kb. **(D)** Metaplots showing H3K27me3 enrichment across a 30 kb region centered on PAX3/7-bound CRMs with significant differential H3K27me3 deposition between WT and DKO organoids. Data are shown for WT and DKO organoids at days 3 and 6 of differentiation. **(E)** UCSC Genome Browser screenshots showing normalised RNA-seq (RNA), FLAG CUT&Tag, ATAC-seq (chromatin opening), or H3K4me2 CUT&Tag read distributions in wild-type (WT), FLAG-PAX3 (PAX3), FLAG-PAX7 (PAX7) expressing, and Pax3; Pax7 double-knockout (DKO) organoids, treated or untreated with BMP4, at days 3, 5, or 6 (D3, D5, D6) of differentiation. Signals are shown in CPM. Coordinates: chr10:19355317–19416668, scale bar: 10 kb. **(F, G)** Metaplots showing ATAC-seq **(F)** and H3K4me2 CUT&Tag **(G)** signal enrichment over a 3 kb region centered on PAX3/7-bound CRMs displaying significant differences in chromatin accessibility or H3K4me2 deposition between CTRL and DKO organoids at day 6 of differentiation. Data are shown for WT and DKO organoids at days 3 and 6 of differentiation. **(H)** Enrichment analysis of PAX3/7-activated and -repressed genes (from Fig 4) among genes located near PAX3/7-bound CRMs displaying significant differential ATAC-seq signals (blue), H3K4me2 deposition (green), or H3K27me3 deposition (dark red) between CTL and DKO organoids at day 6 of differentiation (bars: −log$_{10}$(*p*-value)). The data underlying this figure can be found in S2 Table.

BMP4-specific CRMs were also enriched for PrD–HD and PrD motifs, while untreated-specific CRMs had notably lower PAX motif enrichment, suggesting potential indirect recruitment in the absence of BMP4 at some sites (S5D Fig).

We next asked whether PAX3/7 recruitment might involve cooperation with SMAD1/5/9, the canonical effectors of BMP signaling. To test this, we analyzed the enrichment of five well-characterized SMAD-binding motifs—derived from ChIP-seq and EMSA datasets [51–55]—within PAX3/7-bound CRMs. None of the motifs were significantly enriched across the set of PAX3/7-bound CRMs (adjusted $p \geq 1e - 6$; S5E Fig). To benchmark motif detectability, we analyzed SMAD1/5/9-bound regions identified by ChIP-seq in neural stem cells (NSCs) from E13.5 mouse brains [56] a distinct but related system. In this dataset, four of the five SMAD motifs were strongly enriched (adjusted $p < 1e - 10$), confirming both the validity of the motifs and the sensitivity of our analysis. Notably, some SMAD-bound regions in NSCs were located near genes such as *Id1*, *Wnt1*, *Msx1*, and *Atoh1*—targets also regulated by PAX3/7 and BMP signaling in spinal organoids (S5G Fig). However, these regions showed minimal spatial overlap with PAX3/7-bound CRMs (S5F and S5G Fig). Similarly, across the full set of PAX3/7 CRMs, there was no evidence of SMAD1/5/9 binding in NSCs (S5F Fig), indicating that SMADs and PAX3/7 do not co-occupy shared sites. Thus, while PAX3/7 and SMAD1/5/9 may regulate overlapping gene sets, they likely act through distinct CRMs, arguing against direct cooperation at the level of DNA binding.

Finally, we expanded our motif analysis to explore potential cooperation with other transcription factors (S5H Fig). Motifs for homeodomain, SOX, and SP/KLF zinc-finger TF families were significantly enriched across all PAX3/7-bound CRMs (S5H Fig), consistent with known co-regulatory TF interactions during neural development [46,57–63]. Notably, ZIC TF motifs were specifically enriched in BMP4-specific CRMs (S5Hi Fig). Given the dorsal restriction of ZIC TFs in the spinal cord [64], such PAX-ZIC interactions may support locus-specific recruitment under BMP4 conditions.

We then characterized the chromatin state of PAX3/7-bound CRMs during differentiation and assessed whether this state depended on PAX activity. We performed CUT&Tag profiling mapped H3K27me3 (Polycomb-mediated repression) and H3K4me2 (MLL-associated activation) [65,66] in control and mutant organoids. Additionally, ATAC-seq identified accessible genomic regions (Fig 5C–5H).

Among PAX3/7-bound CRMs, approximately 100 were marked by H3K27me3 in both BMP4-treated and untreated organoids, with half shared between conditions (Figs 5C, 5D and S5I). This mark was dynamically deposited during differentiation—absent at day 3, detectable by day 5, and more pronounced by day 6 (Figs 5C, 5D, S6Ai and S6Aii). Notably, H3K27me3 levels dropped at over 60 of these CRMs in absence of PAX3 and PAX7, regardless of BMP4 treatment (Fig 5D). Nearby genes were enriched for PAX3/7-repressed targets, reinforcing their role in transcriptional repression (Fig 5H). Several CRMs near *Prdm12* gene illustrated this PAX-dependent regulation (Fig 5C). A partial reduction in H3K27me3 levels was observed in both *Pax3$^{-/-}$* and *Pax7$^{-/-}$* organoids, but the effect was markedly more pronounced in double knockout (DKO) mutants (S6Bi–S6Bii and S6Di Fig). In untreated conditions, the reduction in H3K27me3 was

comparable between Pax3⁻/⁻ and Pax7⁻/⁻ organoids (S6Bii and S6Di Fig). However, under BMP4 treatment, the loss of PAX3 resulted in a slightly stronger decrease in H3K27me3 than the loss of PAX7 (S6Bi and S6Di Fig).

Unlike H3K27me3 deposition, which was restricted to a small subset of PAX3/7-bound CRMs, chromatin opening (ATAC-seq) was observed over half of these sites, regardless of culture conditions (S5I Fig). A subset of these CRMs required PAX activity for opening (Figs 5E, 5F and S5I), with 580 CRMs gaining accessibility in BMP4-treated organoids but only 85 in untreated ones—highlighting a strong BMP-dependent effect (Fig 5E and 5F). The transcriptional relevance of these PAX3/7-dependent open CRMs is supported by the enrichment of PAX-activated genes near them (Fig 5H). Among BMP4-responsive CRMs, PAX3/7 opened both BMP4-specific and shared CRMs in nearly equal numbers (S5I Fig). However, 70% of BMP4-specific CRMs required PAX3/7 for opening, compared to just 14% of shared CRMs, further indicating a stronger dependence on PAX activity under BMP stimulation (S5I Fig). More than half of PAX-dependent open CRMs were initially closed at day 3 but progressively opened during differentiation (Figs 5E, 5F, S5I and S6Aiii–S6Aiv), consistent with a chromatin remodeling role akin to pioneer factors described in pituitary [67]. Loss of PAX3 or PAX7 alone caused only mild reductions in accessibility, while their combined loss had a much stronger effect (S6Biii, S6Biv and S6Dii Fig), pointing to a high degree of functional convergence between the two factors, even under BMP4 conditions.

As for H3K4me2, a histone mark associated with accessible and transcriptionally permissive chromatin, 272 PAX3/7-bound CRMs were decorated with this modification—nearly all (265) in BMP4-treated organoids (Figs 5E, 5G, S5I, S6Av and S6Avi). Deposition of H3K4me2 at these sites was largely dependent on PAX activity, as levels were markedly reduced in Pax3;Pax7 double knockout (DKO) organoids compared to controls (Fig 5G). Genes near H3K4me2-marked regions were predominantly PAX-activated, reinforcing the role of these CRMs in PAX-mediated gene activation (Fig 5H). Similar to ATAC-seq results, loss of PAX3 or PAX7 alone caused only mild reductions in H3K4me2 levels, whereas their combined loss had a substantially stronger effect (S6Bv, S6Bvi, S6C and S6Diii Fig). In BMP4-treated organoids, H3K4me2 levels were similarly reduced in Pax3⁻/⁻ and Pax7⁻/⁻ organoids at most CRMs—for example, at the *Olig3* locus (S6Diii and S6Ci Fig). However, a small subset of CRMs, including those near *Msx1*, *Atoh1*, and *Gdf7*, showed a more pronounced reduction in the absence of PAX3 alone, but not PAX7 (S6Cii–S6Civ Fig). These findings further support the idea that PAX3 and PAX7 act cooperatively and largely redundantly to regulate enhancer activation and chromatin priming during dorsal neural fate specification.

These findings underscore the pivotal role of PAX proteins in shaping the chromatin landscape of CRMs in dorsal spinal progenitors. They regulate H3K27me3 deposition near repressed genes, both in dp1–dp3 and dp4–dp6 progenitors. In dp1–dp3 progenitors, PAX3 and PAX7 also promote chromatin accessibility at CRMs near activated genes, opening closed regions, maintaining open states, and facilitating enhancer-associated activation marks. Notably, these functions are largely redundant between the two factors, as phenotypes in single mutants are mild compared to the double knockout. Thus, PAX proteins exhibit dual transcriptional activity, modulated by progenitor identity and target CRM sequences.

## PAX3/7-bound silencers and enhancers shape DV gene expression restriction

We finally investigated the implication of PAX3/7-bound CRMs in DV patterning of the spinal cord (Fig 6). First, we analyzed the enrichment of DV patterning genes near PAX3/7-bound CRMs whose regulatory state depended on PAX activity (Fig 6A). PAX3/7-bound CRMs were classified into three categories based on PAX-dependent modifications: those marked by H3K27me3 deposition, H3K4me2 deposition, or chromatin opening. Genes located near opened and/or H4K4me2-marked CRMs were enriched for dp1–dp3 regulators, but only in BMP4-treated organoids (Fig 6A). Conversely, genes near H3K27me3 marked CRMs were strongly associated with ventral V0 and V1 IN fate, regardless of BMP4 treatment status (Fig 6A). Notably, dI4–dI6 regulators showed no significant enrichment near any of the PAX3/7-bound CRMs. These findings highlight a functional segregation of PAX dual activity: repression targets ventral gene CRMs, while activation acts on dp1–dp3 CRMs.

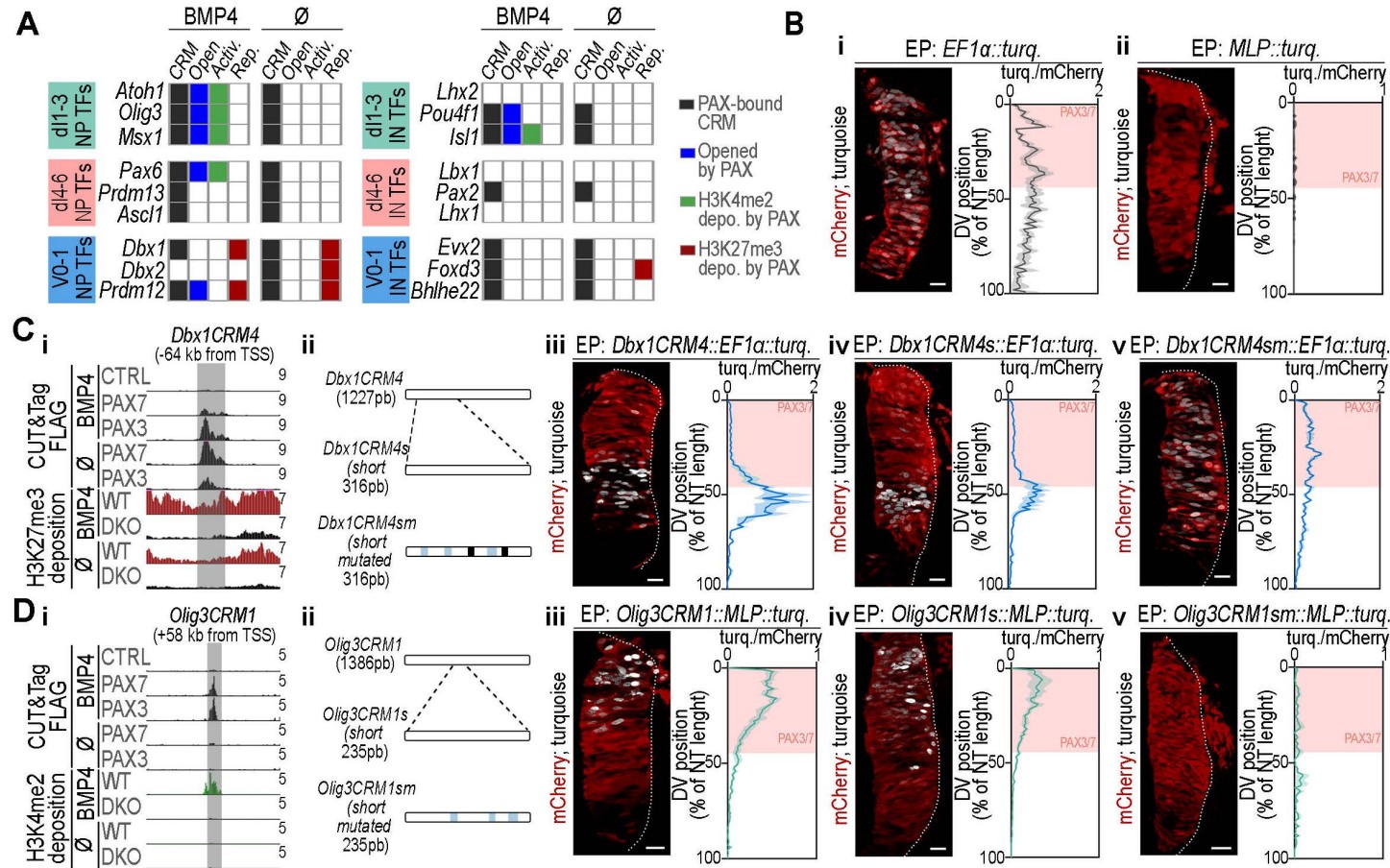

**Fig 6. Silencer or enhancer activity of PAX3/7-bound CRMs nearby DV patterning genes. (A)** Table indicating the presence of PAX3/7-bound CRMs (black) near DV patterning TFs driving dl1–dl3, dl4–dl6, and V0/V1 INs differentiation, along with their chromatin state: opened by PAX (Open, blue), PAX-mediated H3K4me2 deposition (Activ. green), or PAX -mediated H3K27me3 deposition (Rep. dark red). **(B)** Fluorescence of mCherry (red) and Turquoise (white) on transverse sections of the chick neural tube, 24 h post-electroporation with the reporter plasmids from *EF1α* (**i**) or *MLP* (**ii**) promoters' activity. **(C, D) Panel i:** UCSC Genome Browser screenshots showing normalised FLAG (black tracks) CUT&Tag read distributions in control (CTRL), FLAG-PAX3 (PAX3), FLAG-PAX7 (PAX7) at day 5 of differentiation and normalized H3K27me3 (dark red tracks) or H3K4me2 (green tracks) CUT&Tag read distributions in wild type (WT) and Pax3; Pax7 double-knockout (DKO) organoids at day 6 of differentiation, either treated or not BMP4. Read scales are shown in CPM. Grey bands highlight the position of *Dbx1CRM4 chr7:49697106-49701929* (C) and *Olig3CRM1* CRMs (*chr10:19412350-19416446* (D). **Panel ii** Position of the shortened (s), or shortened and mutated (sm) version of either *Dbx1CRM4* and *Olig3CRM1*, with the following color scheme: black for PrD, light blue for PrD-HD. **Panels iii–v: Images:** Fluorescence of mCherry (red) and Turquoise (white) on transverse sections of the chick neural tube, 24 h post-electroporation with pCAG-mCherry and the indicated constructs. **Graphs:** Quantification of the ratio between Turquoise and mCherry signal intensities along the dorsoventral (DV) axis of the neural tube (NT) expressed as a percentage of NT length (bar plots represent mean ± s.e.m.; *n* ≥ 4 embryos). Salmon-colored rectangles indicate the positions of PAX3/7-expressing progenitors. In all images, white dotted lines outline neural tubes. Scale bars: 50 μm. The data underlying this figure can be found in S2 and S7 Tables.

We then assessed the transcriptional potential of selected CRMs and their spatial activity along the DV axis of the neural tube in chick embryos (Figs 6B–6D and S7). Ten PAX-dependent H3K27me3-marked CRMs were cloned upstream of the *EF1α* promoter, which normally drives uniform *turquoise* transgene expression along the DV axis (Fig 6Bi). Among them, *Dbx1CRM4,* located near *Dbx1*, repressed *EF1α* promoter activity throughout the PAX3/7 domain, except in a ventral row likely corresponding to dp6 progenitors (Figs 6Ci, 6Ciii and S7A). This indicates that *Dbx1CRM4* silences transcription in dp1–dp5 progenitors. To further investigate PAX involvement in this silencing activity, we shortened *Dbx1CRM4* and mutated its inner PAX binding sites (Fig 6Cii and S3 Table). The shortened *Dbx1CRM4* (*Dbx1CRM4s*)

retained its ability to repress *EF1α* promoter activity dorsally (Fig 6Civ). However, its mutated version (*Dbx1CRM4sm*) failed to do so, resulting in Turquoise expression along the neural tube without any DV restriction (Fig 6Cv).

Twenty PAX3/7-bound CRMs requiring PAX for chromatin opening and H3K4me2 deposition were cloned upstream of the minimal promoter (*MLP*), which is transcriptionally inactive on its own (Fig 6Bii). One-third of these CRMs activated *MLP* promoter activity in PAX3/7$^+$ progenitors, including CRMs in the vicinity of *Olig3 (Olig3CRM1)*, *Msx1* (*Msx1CRM3*), and *Slit1* (*Slit1CRM1*). Strikingly, their enhancer activity was restricted to the most dorsal region of PAX3/7$^+$ domain, corresponding to dp1 to dp3 progenitors, indicating spatially confined activation (Figs 6Diii, S7A, S7Biii and S7Ciii). Shortened versions of these CRMs retained *MLP* activation, although some of them were extended ventrally, suggesting the loss of sequences restricting DV boundaries (Fig 6Dii, 6Div, S7Bii, S7Biv, S7Cii and S7Civ). Furthermore, mutating PAX motifs in these truncated CRMs completely abolished transcriptional activity (Figs 6Dii, 6Dv, S7Bii, S7Bv, S7Cii and S7Cv).

These findings demonstrate that PAX-modulated CRMs function as silencers or enhancers, with distinct regulatory roles along the spinal cord DV axis. PAX3/7-bound enhancers are active specifically in dp1–dp3 progenitors, mirroring the *P34::tk::LacZ* reporter gradient (Figs 6D and 1B, S7). In contrast, silencing extends more broadly within the PAX domain, coexisting with activation in dp1–dp3 progenitors.

## Discussion

Our study reveals that the paralogous TFs PAX3 and PAX7 play a pivotal role in generating spinal cord cellular diversity by guiding dorsal spinal progenitors (dp1 to dp6) along distinct differentiation trajectories. Their dual transcriptional activity—encompassing both activation and repression—is central to this process. Key parameters underlying the implementation of this dual activity were identified as critical for spinal cell fate patterning. First, like other TFs, such as FOXD3 in neural crest cells (NCCs) or reprogramming factors OCT4, SOX2, and KLF4 (OSK) [68,69], PAX3/7-mediated activation and repression govern distinct differentiation programs: repression restricts ventral fates (p0–p1), while activation drives dorsal progenitor (dp1–dp3) differentiation. This segregation occurs at the genomic level, where PAX3/7-regulated enhancers and silencers occupy distinct genomic regions. Second, unlike FOXD3 and reprogramming factors—whose repressive and activating functions are temporally separated [68,69]—PAX3/7 activities can coexist within the same cell type. FOXD3, for instance, first maintains NCC multipotency before shifting to a repressive role that commits cells to differentiation [69]. In contrast, PAX3/7-mediated activation and repression operate simultaneously within dp1–dp3 progenitors, as demonstrated by the co-occurrence of *Dbx1CRM4* silencing and *Olig3CRM1* enhancer activity. Finally, the DV specificity of PAX dual activity is dictated by its cellular context, particularly BMP exposure. Ultimately, there is a dorsoventral specificity in the modulation of PAX dual activity, dictated by the dependence of PAX-mediated activation on the cellular context established by BMP signaling. Since BMP signaling is spatially restricted to dorsal regions, PAX activation is likewise confined, thereby providing a spatial input to the regulation of its target genes. Within this framework, the following sections examine the distinct contributions of each paralog to this process, as well as the mechanisms through which their gene activation and repression activities are implemented and how they influence dorsal spinal cord patterning.

Although previous studies have suggested that PAX3 and PAX7 may have divergent functions—possibly due to differences in DNA-binding preferences, with PAX3 favoring the paired domain and PAX7 the homeodomain [70,71]— our findings provide little evidence to support such divergence during spinal cord development. The only notable distinction concerns the specification of dI1 interneurons. In spinal cord organoids, *Pax3* deletion leads to a marked reduction in LHX2$^+$ dI1 neurons, whereas *Pax7* deletion causes a slight increase. In embryos, PAX3-FOXO1 promotes dI1 generation, while PAX7-FOXO1 shows no consistent effect. While these findings might suggest a functional divergence, it appears to be only partial, as *Pax7* loss further reduces dI1 neuron generation in *Pax3*-deficient backgrounds, both in vitro and in vivo.

Phenotypic differences between PAX3 and PAX7 mutants, observed both in vitro and in vivo, likely reflect differences in their expression dynamics. For example, in BMP-treated conditions, PAX3 mutants exhibit stronger phenotypes than

PAX7 mutants, but in both settings, PAX3 is expressed earlier and at higher levels than PAX7 [6–9]. In contrast, under BMP-free conditions promoting dp4–dp6 differentiation, PAX3 and PAX7 show more similar expression patterns and mutant phenotypes, further suggesting that phenotypic differences may result from expression level differences rather than intrinsic functional divergence.

The functional convergence and partial redundancy between the two factors are supported by the observation that many phenotypes caused by the loss of one factor are partially rescued by the other, and that simultaneous loss of both paralogs leads to more severe defects. In line with this, *Pax3*<sup>+/PAX3-FOXO1</sup> and *Pax3*<sup>+/PAX7-FOXO1</sup> embryos exhibit highly similar phenotypes, reinforcing the idea that the activator potential of both proteins can similarly modulate dorsal progenitor fate. Finally, our CUT&Tag experiments confirm extensive overlap in the genomic binding profiles of PAX3 and PAX7, along with shared effects on chromatin state, further underscoring their molecular redundancy.

The generic activity of PAX3/7 across their expression domain is transcriptional repression, echoing the critical role of repression in TF-mediated spinal cord patterning [4]. However, this repression does not universally target CRMs near genes regulating all alternative fates to PAX-expressing cells, as observed for ventral TFs [61,72]. For example, genes specifying SHH-dependent ventral identities (p2, p3, pMN) remained unaffected by the PAX3 and PAX7 modulation. Instead, PAX3/7-bound silencers are primarily associated with p0 and p1 progenitor regulators and their V0/V1 neuronal derivatives. Furthermore, PAX3 is transiently expressed in p0 and p1 progenitors [7], suggesting that repression may be lifted during the differentiation, potentially influencing its timing.

The specification of dp4–dp6 and p0–p1 identities has long been debated, as classical neural tube signaling pathways, such as Shh and BMPs, do not fully explain their establishment [7,48,73–75]. These intermediate spinal cord regions are only transiently exposed to noisy dorsal (BMP) and ventral (Shh) morphogen gradients [73]. Moreover, exposure to these signals often inhibits intermediate fate specification, suggesting these identities emerge as default states of spinal progenitors [73–75]. Consistently, in our organoid differentiation experiments, dp4–dp6 and p0–p1 identities arise in the absence of factors modulating BMP or Shh signaling ( [27] and this study). Our findings reveal that PAX-mediated repression imposes a choice between these identities. In the absence of PAX factors, cell fate specification shifts toward p0–p1 states ([10,11] and this study), even in the presence of BMPs. Among PAX-bound silencers, several are located near *Dbx1* and *Prdm12*, key determinants of these ventral identities, whose deregulation in the absence of PAX could explain the ventralisation of dorsal identities [47,48].

PAX3/7-mediated transcriptional repression is particularly striking, as PAX proteins are predominantly recognized for their activator role in cellular differentiation across embryonic tissues [16]. However, our findings may extend to these systems, as studies on a limited number of CRMs and co-repressors have suggested a repressive function for PAX proteins [76,77]. In the spinal cord, our results suggest that this repression involves modulating H3K27me3 deposition, a process typically governed by the Polycomb complexes. PAX2 is known to recruit these complexes through Groucho/TLE co-repressors, binding through a conserved octapeptide domain shared with PAX3 and PAX7 [78,79]. Several lines of evidence support this mechanism in PAX3- and PAX7-mediated repression: (i) PAX7 biochemically interacts with TLE4 [15]. (ii) Deleting the octapeptide domain enhances PAX3 activator potential in myoblasts [80]. (iii) Inhibiting TLE activity in the neural tube of chicken embryos leads to ectopic expression of genes, such as *Dbx1*, normally repressed by PAX [15]. Moreover, in light of a study on the Dorsal protein in *Drosophila*, we propose that such a PAX-TLE repression mechanism may also enable simultaneous activation and repression in dp1–dp3 cells [81]. Like Dorsal, PAX octapeptide diverges from the canonical high-affinity Groucho/TLE-binding sequence, likely reducing its affinity for TLE [15,78]. This lower affinity may make PAX-TLE interactions highly context-dependent, fine-tuning PAX-mediated transcriptional regulation [81].

In contrast to PAX-mediated repression, the transcriptional activation mechanisms of PAX proteins in dp1 and dp3 progenitors align closely with previous findings [17–21,23]. In the spinal cord, PAX proteins are required for opening or maintaining chromatin accessibility at CRMs and, in some cases, for depositing the enhancer mark H3K4me2. This mirrors studies on PAX7 recruitment to CRMs, where it functions as a pioneer factor, recruiting an H3K9me2 demethylase and the

MLL methyltransferase complex to modify H3K4 [18]. Following cell division, which enables the dissociation of PAX3/7-bound CRMs from the nuclear lamina, PAX7 can recruit SWI-SNF nucleosome remodelers and the coactivator P300.

The restriction of PAX3- and PAX7-mediated transcriptional activation to BMP-exposed progenitors is intriguing. While the exact mechanisms remain to be uncovered, several promising hypotheses emerge from our study. A direct interaction between PAX proteins and SMADs, the BMP transcriptional effectors, seems unlikely, as PAX3/7 activity persists for up to four days—far exceeding the transient BMP peak, which declines within an hour [27]. Moreover, PAX3/7-bound CRMs show no enrichment for SMAD-binding motifs. The presence of CRMs specific to dp1–dp3 progenitors suggests that BMP signaling may act indirectly, possibly by inducing factors that facilitate PAX recruitment. Comparisons between SMAD binding in anterior neural stem cells and PAX binding in spinal organoids suggest that these regulators may act on distinct CRMs located near the same target genes, potentially functioning in regulatory cohorts rather than through shared elements. Alternatively, the enrichment of ZIC motifs in these CRMs suggests a plausible BMP-specific PAX-ZIC partnership, further supported by the exclusive expression of ZIC1, ZIC2, and ZIC5 in BMP-exposed progenitors [64], and the demonstrated interaction between PAX3 and ZIC2 in regulating a *Myf5* CRM [82]. Alternatively, BMP effectors may modulate PAX activity on already bound CRMs rather than influencing its recruitment. Supporting this, nearly half of PAX-activated CRMs in BMP-treated conditions were PAX3/7-bound regardless of BMP exposure and the *P34::tk::LacZ* reporter transgene, containing only PAX-binding motifs, is restricted dorsally. Post-translational modifications (e.g., acetylation [14], SUMOylation [83], phosphorylation [84]) or alternative splicing could regulate PAX activity [16,85,86], and so could the recruitment of specific cofactors such as ZIC1 [82] or Lef/TCF [87].

## Materials and methods

### Mouse lines

Mice carrying the *Pax3* knock-in *GFP* null allele (*PAX3^GFP^*), conditional PAX3-FOXO1 (*PAX3^PAX3-FOXO1^*), Pax3-EnR (*PAX3^Pax3-EnR^*), and *Pax7* knock-in *LacZ* null (*PAX7^LacZ^*) alleles and the *P34::tk::LacZ* reporter transgene have been previously described [12,24,33,34]. The *PAX3-FOXO1* and *Pax3-EnR* alleles were expressed from the *Pax3* locus upon activation of Cre recombinase, driven by the zygote-specific *PGK* enhancer [88]. We have also generated mice carrying a *Pax3* knock-in conditional PAX7-FOXO1 (*PAX3^PAX7-FOXO1^*) mouse line. The targeting construct is derived from a previously reported design [24]. Cloning details are available upon request. Briefly, the *Pax3^EGFP(Pax7-FOXO1A-IRES-nLacZ)^* allele contains 2.4 kb of the 5′ genomic region of Pax3, excluding the coding sequence of exon 1, and 4 kb of the 3′ sequence encompassing exons 2–4. This genomic fragment is flanked by a floxed *GFP–FRT–Puromycin* cassette, followed by 2.5 kb of *PAX7-FOXO1*, then an *IRES–nLacZ* cassette, and an FRT site. Additionally, a PGK-DTA cassette encoding the A subunit of the diphtheria toxin gene [89] was inserted at the 5′ end of the construct for negative selection in ES cells. The targeting vector was electroporated into CK35 ES cells [90]. Recombinant ES cells were selected and screened by Southern blot analysis using EcoRV digestion and a 5′-flanking probe. Positive clones were further verified with 3′ external and internal probes (details are available upon request). Targeted ES cells were recovered at a frequency of 0.5–1% and injected into blastocysts to generate chimeric mice. Germline transmission was confirmed by PCR. Cre and FLP transgenic mice, described previously [88,91], were used to generate the *Pax3^PAX7-FOXO1-IRES-nLacZ^* allele (abbreviated *Pax3^PAX7-FOXO1^*) by removing the EGFP sequence (via Cre) and excising the selectable Puromycin cassette (via FLP). All experiments were conducted in accordance with the guidelines of the Institutional Animal Care and Use Committee of Université Paris Cité (CEB11.2025), under authorization number 01416.02 from the French Ministry of Research, for a project entitled: "Study of the gene regulatory network controlling dorsal spinal cord development".

### mESC cell lines maintenance and differentiation in organoids

All mouse ESC lines were maintained on mitotically inactive primary mouse embryo fibroblasts in D-MEM Glutamax supplemented with 15% ESC-qualified fetal bovine serum (Millipore), L-glutamine, non-essential amino acid,

nucleosides, 0.1 mM β-mercaptoethanol (Life technologies) and 1,000 U.ml⁻¹ leukemia inhibitory factor (CELL GS). Dorsal spinal organoids were generated from mESC according to previously published protocol [27] (Fig 3A). In brief, cells were trypsinized and placed twice onto gelatinized tissue culture plates to remove feeders. From $5 \times 10^5$ cells.ml⁻¹ to $1 \times 10^6$ cells.ml⁻¹ were placed in ultra-low attachment petri dishes (Corning) and in Advanced Dulbecco's Modified Eagle/F12 and Neurobasal media (1:1, Life technologies) supplemented with 1× B27 devoid of Vitamin A and 1× N2 (Life technologies), 2 mM L-glutamine (Life technologies), 0.1 mM β-mercaptoethanol, 100 U/mL penicillin and 100 U/mL streptomycin (Life technologies). At this concentration of cells, small EB were formed from day 1 of differentiation, and grown as such up to day 7 of differentiation; medium was changed every day from day 2 onwards. Chemical drugs to inhibit or activate key develop mental signaling pathways were used at the following concentrations: 3 µM CHIR99021 (Tocris or Axon Medchem) from day 2 to day 3, 10 nM Retinoic Acid (Sigma) from day 2 to day 7, 5 ng.ml⁻¹ BMP4 (R&D) from day 3 to day 4 (Fig 3A).

## Generation of transgenic mESC

The *P34::tk::LacZ* mESC line was derived from mouse blastocysts following a previously published protocol [92]. To create *Pax3⁻/⁻*, *Pax7⁻/⁻*, or *Pax3⁻/⁻; Pax7⁻/⁻* lines, we employed CRISPR-Cas9-mediated deletion of the first two exons and half of the third exon for each gene (this deletion eliminates five ATG codons and creates a frameshift). The guide RNAs used are provided in S4 Table. For *Pax3⁻/⁻*, following passage, $1 \times 10^6$ cells were plated and were co-transfected with *pX459* plasmid enabling transient expression of *Cas9* and the sgRNA for *Pax3* first exon, as well as another *pX459* plasmid expressing the sgRNA for *Pax3* third exon. The plasmid *pSpCas9(BB)-2A-Puro (PX459)* V2.0 was a gift from Feng Zhang (Addgene plasmid # 62,988; http://n2t.net/addgene:62988; RRID: Addgene_62988). Transfection was performed using Lipofectamine 2000 (Life technologies, Carlsbad, CA, USA) according to the manufacturer's instructions. Twenty-four hours later, puromycin (2 µg/ml) was added to the medium. After two days, the cells were diluted 1:100 and replated. Four days later, 96 isolated colonies were picked and split into two portions. After removal of feeders, DNA was extracted using the RedExtract-N-Amp kit, and the clones were genotyped via PCR (S5 Table). Three independent homozygous clones per genotype were selected. The same strategy was applied for *Pax7⁻/⁻*. For *Pax3⁻/⁻; Pax7⁻/⁻* lines, cells were co-transfected with the four plasmids. To generate *Flag-Pax3* cell line, cells were co-transfected with a *pX459* plasmid expressing the sgRNA *Pax3sg385* and a single-stranded donor DNA enabling the integration of *3xFLAG* tag upstream of the first ATG codon of *Pax3*, using the same transfection protocol (S4 Table). After genotyping, only one clone exhibited the correct integration of the tag; this clone was amplified and used for CUT&Tag experiments. The same approach was used to generate the *Flag-Pax7* cell line using sgRNA *Pax7sg613*.

## Reporter constructs and in ovo electroporation

Enhancers or silencers bound by PAX3/7 were amplified by PCR and inserted upstream of the minimal *adenovirus Major Late Promoter* (MLP) and *H2B-Turquoise* for enhancers, or upstream of the *EF1alpha* constitutive promoter and *H2B-Turquoise* for silencers, using the Takara In-Fusion kit. The mutant versions of CRMs were obtained by PCR-mediated directed mutagenesis using the QuickChange Multi Site Directed Mutagenesis Kit from Agilent. Reporter plasmids (1 µg/µl) and a *pCAG* plasmid enabling constitutive expression of the mCherry reporter (0.3 µg/µl) were electroporated into Hamburger and Hamilton (HH) stage 10–11 chick embryos following established protocols [44]. *BRE::MLP::Turquoise* was generated from a previously described multimerized Smad1/5-responsive BREs [30], with the luciferase cassette removed by HindIII/XbaI digestion and replaced with an *H2B–Turquoise* fusion (gift from A. Kicheva). Expression vectors expressing *Alk3ca, Smad6, Pax3-EnR, Pax7-EnR, PAX3-FOXO1,* and *PAX7-FOXO1* were previously described [10,32,93,94] and electroporated at the same stage at a concentration of 1 µg/µl together with the *P34::tk::LacZ* or *BRE::MLP::Turquoise* reporters (0.5 µg/µl). Embryos were dissected at the indicated stage in cold 1× PBS.

## Immunohistochemistry on histological sections

Mouse and chick embryos were fixed in 4% paraformaldehyde for 45 min to 2 h at 4 °C. Fixed embryos were cryoprotected by equilibration in 15% sucrose, embedded in gelatine, cryosectioned at 14 µm, and processed for immunostaining [44]. The fixation, embedding, and cryosectioning of EBs were conducted as previously described [95], and immunolabelling was performed using the same protocol as for embryo sections. Details of the antibodies are provided in S6 Table. Images were carried out using either a Leica TCS SP5 confocal microscope or a Zeiss LSM 980, and images were processed with Adobe Photoshop 7.0 (Adobe Systems, San Jose, CA, USA) or ImageJ v.1.43g (NIH). All scale bars represent 50 µm. In mouse embryos, quantifications were performed at brachial levels on usually more than 3 embryos and on 2–4 transverse sections per embryo.

## Quantification based on immunostaining signals

The number of cells per section of mouse spinal cords was determined using the Cell Counter plugin in ImageJ v.1.43g (NIH). We implemented a batch-wise normalization strategy: for each experimental series, values were scaled such that the average count in heterozygous embryos matched an expected value at E11.5. This approach reduced inter-litter variability while preserving biologically meaningful differences between genotypes. The fluorescence intensity of β-galactosidase, GFP, Turquoise, mCherry and pSMAD1/5/9 in chick and mouse embryos, as well as the evaluation of dorsal–ventral boundary positions, were determined as described in [96]. The number of cells immuno-labelled in organoids was calculated using Cell Profiler (Broad Institute) after nuclei segmentation based on the DAPI fluorescence signal and a defined signal intensity threshold, expressed as a percentage of all detected nuclei. Fluorescence signal intensity levels of PAX3, PAX7, or phosphorylated SMAD1/5/9 per cell were evaluated using CellProfiler, followed by background subtraction. Background values were determined in EBs not treated with BMP4 for phosphorylated SMAD1/5/9 or in EBs generated from *Pax3⁻/⁻; Pax7⁻/⁻* cell lines. Statistical analyses and graphs were performed using GraphPad Prism software. For each graph, the biological replicates (embryos or ESC clones, typically ≥3) and technical replicates (individual sections) were displayed. Non-parametric *t*-tests (Mann–Whitney U test) were used for pairwise comparisons between conditions. *P*-values are denoted as follows: * for $p \leq 0.05$, ** for $p \leq 0.01$, *** for $p \leq 0.001$, and **** for $p \leq 0.0001$. All quantified data can be found in S7 Table.

## RNA extraction, qPCR and sequencing

RNAs were extracted using the NucleoSpin RNA kit (Macherey-Nagel) following manufacturer's instructions, elution was done with RNase-free water and concentrations were measured with DeNovixDS-11-FX series. For RT-quantitative real-time-PCR, cDNA were synthesized using SuperScript IV (Thermo Fischer Scientific), random primers and oligo dT, then SYBR Green I Master (Roche) and Prime Pro 48 real time qPCR system (Techne) were used. PCR primers were designed using Primer-Blast (S8 Table). Levels of expression per gene for a given time point was measured in biological duplicates or triplicates. Gene expression levels were normalised to the levels of *Ywhaz* mRNA in ESCs. For transcriptomic analysis, RNAs were sent to BGI (Hong-Kong) where libraries were prepared and their quality checked using a Bioanalyzer Agilent 2,100. 0.2–0.5 µg was then either sequenced on a DBN sequencing system on paired-end reads 150 base runs (D6 organoids). After demultiplexing, reads were trimmed and mapped on the mm10 mouse genome (BGI).

## Transcriptomic analyses

Pairwise differential analysis of RNA-seq data was performed using Deseq2 [97] (BGI). Differentially expressed genes were defined as those with a mean expression >5 FPKM in at least one group, an absolute fold change >1.5, and a *q*-value < 0.05 after Benjamini–Hochberg correction for multiple-testing were considered to be significant. Volcano plots were generated using ggplot2 [98]. Line plots were generated in GraphPad Prism. Ad-hoc enrichment analyses between

differentially expressed genes, and gene sets defining DV populations (S1 Table) were performed using hypergeometric tests (phyper function on R). The lists of genes coding for DV markers genes (S1 Table) were generated based on literature reviewing and were used to project the effects of the PAXs onto the acquisition of DV identities. Heatmaps of relative expression levels were generated using the plotHeatmap function in R.

## ATAC-Seq, CUT&Tag and sequencing

ATAC-Seq was performed as in [99]. We used 50,000 cells per condition, and a homemade purified pA-TN5 protein. CUT&Tag was performed as in [100], following the procedure described in "Bench top CUT&Tag V.2" available on protocols.io. We used 500,000 cells per sample and the anti-FLAG (Sigma-Aldrich F3165-5MG; 1/50) for CUT&Tag on transcription factors, or 200,000 nuclei that were previously fixed and frozen following the procedure described in [100] with anti-H3K4me2 (Abcam ab176878; 1/100), anti-H3K27me3 (Cell Signaling 9733S; 1/100) primary antibodies; and anti-rabbit IgG (ABIN6923140) or anti-mouse IgG (ABIN6923141) secondary antibodies. We used a homemade purified pA-TN5 protein. Libraries were analyzed using 4,200 TapeStation system (Agilent) and sequenced 100 pb (paired-end) with DNBSEQ-G400 (BGI). These data can be found in S2 Table.

## Bioinformatic analyses

The quality of paired-end read sequences was assessed using FastQC (v0.11.9). Low quality nucleotides and Illumina adapter sequences were removed using Trimmomatic (v0.39) [101] and PCR duplicate reads were removed using BBmap clumpify (v38.87) [102]. Filtered reads were aligned to the mm10 reference genome using Bowtie2 (v2.4.5) [103] and parameters "--local --very-sensitive --no-mixed --dovetail --no-discordant -- phred33 -I 10 -X 700". Genome-wide coverage tracks (bigwigs) were generated using deeptools (v3.5.1) [104] bamCoverage and parameter "--normalizeUsing CPM --blackListFileName blacklisted_regions.fa --ignoreForNormalization chrX chrM chrY --binsize 1", removing blacklisted regions defined by the Kundaje lab. Peaks were called using MACS2 (v 2.2.7.1) [105] with parameters: -f BAMPE -broad -g 1.87e9 and manually curated. Reads coverage on defined CRMs was calculated using VisRseq (v0.9.2) by taking the signal on each CRM, then normalised by library size. Differential analysis on PAX bound peaks was performed using Deseq2 (v1.42.1) [97], allowing the identification of three CRMs categories (abs(fold change) > 1.5; adjusted $p$-value < 0.05). For ATAC-Seq, H3K4me2 and H3K27me3, peaks were called in every condition using the same MACS2 parameters, and for H3K27me3, peaks within 5000 bp of each other were merged. Peaks obtained for ATAC in the different conditions were merged to form a universal of list of peaks, and the same was done for the other chromatin parameters. Subsequently, enrichment was calculated using VisRseq, and differential analyses were performed using DeSeq2, allowing us to define differentially open, activated or repressed regions. Using bedtools intersect between PAX CRM and these marks data, we could identify the chromatin state of the different PAX CRMs. Heatmaps and average plots were generated using deeptools (v3.5.1) [104] computeMatrix followed by plotHeatmap or plotProfile. PCA plots were generated with plotPCA command in R. The association between PAX3/7-bound CRM and gene was performed using GREAT [49]. Heatmaps of relative expression levels were generated using the plotHeatmap function in R. UCSC Genome Browser [106] track data hubs [107] were used to display data over loci of interest.

## Motifs analyses

Position weight matrices (PWMs) for TFs binding sites were taken from the JASPAR 2022 database [108], to which we added PWM created de novo from PAX3, PAX7 or PAX-FOXO1 ChIP-Sequencing data previously published [71], as well as PWM for pSMAD1/5/9 previously published [51–55]. Relative enrichment of those PWM was calculated in PAX3/7 CUT&Tag data using AME [109] from MEME-suite. The logos of the PAX3/7 binding motifs were generated from PWM using ggseqlogo [110].

PLOS Biology

## Supporting information

**S1 Fig. Temporal dynamics of the *P34::tk::LacZ* dorsal–ventral activity gradient and effects of Alk3 and Smad6 overexpression on BMP signaling and PAX3 expression. (A)** Immunostaining for β-galactosidase (βgal; grey/pink/white), GFP (blue, shown on one side of the neural tube only in top panels), and OLIG3 (green) on transverse sections of the brachial spinal cord from spinal cord sections of *P34::tk::LacZ; Pax3GFP/+* embryos at the indicated developmental stages. Lower panels: Magnified views of βgal+ regions. Black dotted lines contour neural tubes. Scale bar: 50 μm. **(B)** Quantification of β-gal, OLIG3, and GFP signal intensity (in arbitrary units, AU) along the DV axis of the PAX3+ domain, expressed as a percentage of domain length (mean ± s.e.m.). **(C)** Quantification of Turquoise/GFP signal ratio in the neural tube of chick embryos 24 h post-electroporation (hpe) with the indicated constructs. Signal is mapped along the DV axis as a percentage of total neural tube length (mean ± s.e.m.). **(D)** Quantification of the effects of ALK3ca and SMAD6 overexpression on PAX3 expression levels and spatial distribution in the chick neural tube, 24 hpe. **Top graph**: Ratio of PAX3 signal intensity in electroporated (E) cells relative to non-electroporated (NE) cells within the PAX3-positive domain. **Bottom graph**: Ratio comparing the position of the ventral boundary of the PAX3 expression domain between the E and NE sides of the neural tube. Smad6 had no significant effect, while ALK3ca induced a ventral shift of the PAX3 domain, consistent with previous findings showing ventral expansion of PAX7 expression upon ALK3ca overexpression [1]. The data underlying this figure can be found in S7 Table.
(TIF)

**S2 Fig. V1 interneuron markers following modulation of PAX3 and/or PAX7 transcriptional activity in mouse embryos, and description and validation of the *Pax3*PAX7-FOXO1 mouse allele. (A)** Immunostaining for FOXD3 (red), FOXP2 (green) and GFP (Blue) on transverse sections of E11.5 spinal cords from wild-type, *Pax3; Pax7* double mutants, or embryos expressing PAX3-FOXO1 from the *Pax3* locus. Pale blue arrowheads indicate FOXP2high V1-like INs, while white arrows mark FOXD3+; FOXP2− or low dI2-like INs. Only one half of the dorsal neural tube is shown. "**M**" denotes a mixed population of INs displaying V1 or dI2-like characteristics. **(B)** Schematic representation of the *Pax3* locus targeted with the *PAX7-FOXO1* cDNA cassette. The targeting construct includes 2.4 kb of the 5′ flanking region and 4 kb of the 3′ region (including exons 2–4) of the *Pax3* genomic sequence. The coding region of exon 1 is replaced by an EGFP reporter, preceded by a *loxP* site and followed by an FRT site, a *puromycin-polyA* (Puro-pA) selection marker, a second *loxP* site, and a cassette encoding the human *PAX7-FOXO1* fusion protein. This is followed by an *IRES-nLacZ* cassette, an FRT site, and a polyadenylation signal. A counter-selection cassette encoding the A subunit of diphtheria toxin (DTA) is inserted at the 5′ end of the vector. **(C)** Immunostaining for FOXO1 (green) and PAX3 (pink) on transverse sections of E10.5 wild-type or *PAX3+/PAX7-FOXO1* spinal cords. Square panels show magnified views of the areas indicated by the square in panels *x*. Arrowheads indicate FOXO1+ vessels (ves). In all images, white dotted lines contour neural tubes. Scale bars: 50 μm.
(TIF)

**S3 Fig. PAX3 and PAX7 mRNA and protein expression dynamics during dI1–dI3 and dI4–dI6 INs differentiation in spinal organoids. (A)** Heatmaps showing normalised mean-centered mRNA expression of lineage marker TFs or a signaling ligand (*Fgf5*) during spinal organoid differentiation, with or without BMP4 (mean expression relative to *Ywhaz* over 3 independent differentiations of WT mESCs, color-coded from blue (lower levels) to yellow (higher levels)). **(B)** mRNA expression levels of *Pax7* and *Pax3* relative to *Ywhaz* during WT spinal organoid differentiation with (green) or without BMP4 (pink salmon) (mean ± s.e.m over 3 independent differentiations). **(C)** Immunodetection of PAX3 and PAX7 (white) and DAPI-stained nuclei (blue) on sections of day5 organoids grown with BMP4, derived from each of *Pax3−/−*, *Pax7−/−* and *Pax3; Pax7* double knockout (DKO, *Pax3−/−; Pax7−/−*) mESC clones used in the study. **(D)** Quantification of PAX7 **(i)** and PAX3 **(ii)** expression levels in positive cells within day 5 organoids (violin plots: values per cell from organoids across 3 independent differentiations; bars: mean). **(E)** Immunodetection of SOX1 (white) and DAPI-stained nuclei

(blue) on sections of WT and DKO organoids grown with or without BMP4 and quantification of the percentage of SOX1$^+$ cells (dots: values per organoid; dot shapes: clones; bar plots: mean ± s.e.m). **(D)** Immunodetection of HuC/D (white) and DAPI-stained nuclei (blue) on sections of WT and DKO organoids grown with or without BMP4, and quantification of the percentage of HuC/D$^+$ cells (dots: values per organoid; dot shapes: clones; bar plots: mean ± s.e.m; Mann–Whitney U test: *$p < 0.05$; ****$p < 0.0001$). In all images, white dotted lines contour organoids. Scale bars: 50 μm. The data underlying this figure can be found in S7 Table.
(TIF)

**S4 Fig. V1 and V0 INs ectopic generation and BMP signaling in organoids lacking PAX3 and PAX7.** Immunostaining for POU4F1 (red) and FOXD3 (green) **(A)**, and PAX2 (red) and LBX1 (green) **(B)** sections of day 7 organoids with the indicated genotypes, treated or not with BMP4. **Graphs:** Quantification of the percentage of FOXD3$^+$;POU4F1$^+$ dI2 INs and FOXD3$^+$;POU4F1$^-$ V1 INs within the FOXD3$^+$ population, as well as the percentage of FOXD3$^+$;POU4F1$^-$ V1 INs, of PAX2$^+$;LBX1$^+$ dI4/6 INs and of PAX2$^+$;LBX1$^-$ V0–V1 INs relative to all cells (dots: values per transverse section, dots shapes: independent clones; bar plots: mean ± s.e.m; Mann–Whitney U test: *$p < 0.05$; **$p < 0.01$; ***$p < 0.001$; ****$p < 0.0001$). **(C) Images:** Immunostaining for phosphorylated form of SMAD1/5/9 (P-SMAD1/5/9)(white) and DAPI-stained nuclei on sections of day 3 organoids with the indicated genotypes, grown without or with BMP4 for 1 h. **Graphs:** Quantification of the percentage of P-SMAD1/5/9$^+$ cells (dots: values per transverse section, dots shapes: independent organoids; bar plots: mean ± s.e.m) and nuclei levels of P-SMAD1/5/9 in these organoids (dots: values per nuclei; bar: median). In pictures, white dotted lines contour organoids, and scale bars represent 50 μm. The data underlying this figure can be found in S7 Table.
(TIF)

**S5 Fig. Genome position, DNA-binding motifs composition and chromatin state dynamics of PAX3/7-bound CRMs. (A)** Distribution of PAX3/7-bound CRMs in functional genomic regions (expressed as a percentage). TSS: Transcriptional Start Site. **(B)** Enrichment of differentially expressed genes upon loss of PAX3 and PAX7, genes activated or repressed by PAX3 and PAX7 and a random set of genes (CTRL) among genes located near PAX3/7-bound CRMs (bars: −$\log_{10}$($p$-value)). **(C)** Logos of position weight matrices of DNA-binding motifs for the PAX3/7 paired (PrD), the PAX3/7 paired and homeodomain (PrD-HD), and a dimer of PAX3/7 homeodomains. **(D)** Enrichment of the three kinds of PAX3/7 DNA-binding motifs in PAX3/7-bound CRMs specific to BMP-treated organoids (BMP spe.), common to treated and untreated organoids (Common), and unique to untreated organoids (∅ spe.) (bars: −$\log_{10}$($p$-value)). **(E)** Motif matrices enriched at pSMAD1/5/9 genomic binding sites identified in the studies listed on the left. Enrichment scores are shown for each motif in either PAX3/7-bound CRMs we identified or pSMAD1/5/9-bound CRMs identified in (2). **(F)** Heatmaps of normalized FLAG CUT&Tag signals at all PAX3/7-bound CRMs identified in organoids expressing FLAG-PAX3 (PAX3), FLAG-PAX7 (PAX7), or control (CTRL), treated with BMP4 at day 5 of differentiation; signals shown across a 1.5 kb window centered on the CRM midpoint. Heatmaps of normalized pSMAD1/5/9 ChIP-seq signals at PAX3/7-bound CRMs (identified by (2)) in neural stem cells (NSCs) treated or not with BMP4. **(G)** UCSC Genome Browser screenshots showing normalized FLAG CUT&Tag read distributions in wild-type (CTRL), FLAG-PAX3 (PAX3), and FLAG-PAX7 (PAX7) organoids, with or without BMP4 treatment, at day 5 of differentiation. Also shown are normalized pSMAD1/5/9 ChIP-seq signals (identified by (2)) in neural stem cells (NSCs) treated or not with BMP4. Signal scales are in CPM (counts per million). Coordinates: *chr2:152731031–152742630* (**i**, scale bar: 3 kb), *chr15:98788158–98832159* (**ii**, scale bar: 6 kb), *chr15:98788158–98832159* (**iii**, scale bar: 25 kb), *chr6:64676194–65048197* (**iv**, scale bar: 50 kb). PAX3/7 recruitment sites highlighted in grey; pSMAD1/5/9 sites in orange. **(H)** Enrichment of JASPAR database TFs and the three PAX3/7 DNA-binding motifs in PAX3/7-bound CRMs specific to BMP-treated organoids (**i**), common to treated and untreated organoids (**ii**), and specific to untreated organoids (**iii**) (the top 50 motifs per CRM subtype are shown with their rank on the *x*-axis and their enrichment score

($-\log_{10}$($p$-value)) on the $y$-axis. **(I)** Categorization of PAX3/7-bound CRMs subtypes depending on their chromatin state (only open (blue rectangle), open and marked with H3K4me2 (green rectangle) or H3K27me3 (burgundy rectangle), or harboring none of these chromatin marks (grey rectangle)), on whether this state is dependent on PAX activity (solid rectangle: dependent on PAX activity; dashed rectangle: independent of PAX activity), and on whether their opening is acquired between day 3 and day 5 of differentiation (open during differentiation) (black outline). The data underlying this figure can be found in S2 Table.
(TIF)

**S6 Fig. Chromatin state dynamics of PAX3/7-bound CRMs during the course of differentiation and in the absence of PAX3 and/or PAX7. (A)** Heatmaps of normalised H3K27me3 (**i, ii**), ATAC-seq (**iii, iv**, open chromatin), and H3K4me2 (**v, vi**) signals at PAX3/7-bound CRMs in wild-type organoids at the indicated days of differentiation, treated or untreated with BMP4. **(B)** Heatmaps of the same chromatin marks and ATAC-seq signals in day 6 organoids with the indicated genotypes, treated or untreated with BMP4. **(C)** UCSC Genome Browser screenshots of normalized FLAG CUT&Tag read distributions in wild-type (CTRL), FLAG-PAX3 (PAX3), and FLAG-PAX7 (PAX7) organoids, with or without BMP4 treatment, at day 5 of differentiation. Normalized H3K4me2 CUT&Tag signals in day 6 wild-type (WT), *Pax3$^{-/-}$*, *Pax7$^{-/-}$*, and double knockout (DKO) organoids. Signal scales in CPM (counts per million). Coordinates: *chr10:19355021–19452047* (i, scale bar: 10 kb), *chr5:37817008–37902925* (ii, scale bar: 10 kb), *chr6:64674484–64750244* (iii, scale bar: 10 kb), *chr12:8023576–8305562* (iv, scale bar: 50 kb). PAX3/7 recruitment sites highlighted in grey. **(D)** Metaplots showing H3K27me3 enrichment over a 15 kb region upstream and downstream of the centers of PAX3/7-bound CRMs, ATAC-seq mean signal, and H3K4me2 enrichment over a 1.5 kb region upstream and downstream of the centers of PAX3/7-bound CRMs in wild-type (WT) and DKO organoids at day 6 of differentiation. The data underlying this figure can be found in S2 Table.
(TIF)

**S7 Fig. Transcriptional activity of PAX3/7-bound CRMs near DV patterning genes. (A)** Fluorescence of mCherry (red) and Turquoise (white), and immunodetection of PAX7 (green) on transverse sections of the chick neural tube, 24 h post-electroporation with the indicated constructs. Scale bars: 50 μm. **(B, C) Panel i:** UCSC Genome Browser screenshots showing normalised FLAG CUT&Tag read distributions (black tracks) in control (CTRL), FLAG-PAX3 (PAX3), FLAG-PAX7 (PAX7) day 5 organoids and H3K4me2 CUT&Tag read distributions (green tracks) in day6 wild-type (WT) and *Pax3; Pax7* double-knockout (DKO) organoids, treated or untreated with BMP4. Read scales are shown in CPM. Grey bands highlight the genomic position of *Msx1CRM3* (chr5:37895330-37896785) and *Slit1CRM1* (chr19:41716058-41716783). **Panel ii:** Position of the shortened (s), or shortened and mutated (sm) versions of the *Msx1CRM3* and *Slit1CRM1*. Mutated PAX motifs are highlighted in the CRM versions, with the following color scheme: orange for HD-HD, black for PrD, and light blue for PrD-HD. **Panels iii–v:** Fluorescence of mCherry (Red) and Turquoise (white) on transverse sections of the chick neural tube, 24 h post-electroporation with the pCAG-Cherry and the indicated constructs. Scale bars: 50 μm. Quantification of the ratio between Turquoise and mCherry signal intensities along the DV axis of the neural tube (NT), expressed as a percentage of NT length (bar plots represent mean ± s.e.m.; $n > 4$ embryos). Salmon-colored rectangles indicate the positions of PAX3/7-expressing progenitors.
(TIF)

**S1 Table. Differential RNA-seq analyses of day 6 WT, Pax7$^{-/-}$, Pax3$^{-/-}$ and Pax3$^{-/-}$; Pax7$^{-/-}$ organoids.** Excel file containing additional data too large to fit a PDF. **Sheet 1: Column A:** Gene IDs. **Columns B to Y:** Gene expression levels (FPKM) in day 6 wild-type (WT), *Pax7$^{-/-}$*, *Pax3$^{-/-}$* and *Pax3$^{-/-}$; Pax7$^{-/-}$* (DKO) organoids. **Columns Z to AK:** Fold changes and q-values from all pairwise differential analyses performed using DE-Seq2 between the indicated sample pairs. **Sheet 2:** Gene signatures defining spinal pools of neural progenitors and INs.
(XLSX)

**S2 Table. PAX3/7-bound CRMs positions and chromatin states.** Excel file containing additional data too large to fit a PDF. **Sheet 1: Columns A–C:** Positions of all PAX3/7-bound CRMs identified using FLAG-CUT&Tag. **Column D:** Categories of CRMs: PAX3/7-bound CRMs specific to BMP-treated organoids (BMP spe.), common to treated and untreated organoids (Common), and unique to untreated organoids ($\varnothing$ spe.). **Column E:** Closest genes to each CRM, identified using the GREAT tool. **Columns F–Y:** Chromatin state of each CRM, defined through pairwise comparisons between WT and DKO lines at different differentiation times. Each column represents a specific category; a value of 1 indicates the CRM state matches the category.
(XLSX)

**S3 Table. Position and sequence of PAX motifs in PAX bound enhancers electroporated in chick embryos.** First square (Olig3CRM): Genomic position of Olig3CRM1; sequence of the shortened version of Olig3CRM1 (Olig3CRM1s), and mutated version of Olig3CRM1s. Second square (Msx1CRM): Genomic position of Msx1CRM3; sequence of the shortened version of Msx1CRM3 (Msx1CRM3s), and mutated version of Msx1CRM3s. Third square (Slit1CRM): Genomic position of Slit1CRM1; sequence of the shortened version of Slit1CRM1 (Slit1CRM1s), and mutated version of Slit1CRM1s. Fourth square (Dbx1CRM): Genomic position of Dbx1CRM4; sequence of the shortened version of Dbx1CRM4 (Dbx1CRM4s), and mutated version of Dbx1CRM4s. All squares: the PAX binding sites are highlighted in grey.
(DOCX)

**S4 Table. Sequences of the guide RNAs (gRNA) used for generating the knock-out of Pax7$^{-/-}$, Pax3$^{-/-}$ and Pax3$^{-/-}$; Pax7$^{-/-}$ mESC lines, as well as the gRNA and donor DNA (ssDonor) for the generation of FLAG tag knock-in in Pax3 or Pax7 loci in mESC.**
(DOCX)

**S5 Table. Genotypes the three Pax7$^{-/-}$, Pax3$^{-/-}$ and Pax3$^{-/-}$; Pax7$^{-/-}$ mESC lines used in the study.** Referenced chromosomal position of mouse Pax3: chr1: 78,197,134 and Pax7: chr4: 139,833,528.
(DOCX)

**S6 Table. List of antibodies.**
(DOCX)

**S7 Table. All data and statistics shown in the paper's graphs.**
(XLSX)

**S8 Table. Primers sequences used for qRT-PCR on cDNA prepared from mESC derived EB and organoids.**
(DOCX)

## Acknowledgments

We sincerely thank C. Birchmeier, T. Müller, and S. Garel for providing antibodies; E. Marti and G. Le Dréau for providing *ALK3CA* and *SMAD6* expression vectors and Anna Kicheva for providing the *BRE::mlp::Turquoise* reporter. We also acknowledge the ImagoSeine core facility at the Institut Jacques Monod, a member of France-BioImaging (ANR-10-INBS-04) and certified by GIS-IBiSA. We are grateful to the biological service staff of the CDTA and Buffon animal housing for help with the mouse colonies. Finally, we are grateful to S. Nedelec and N. Konstantinides for their critical comments on the manuscript.

## Author contributions

**Conceptualization:** Robin Rondon, Claire Dugast-Darzacq, Frédéric Relaix, Pascale Gilardi-Hebenstreit, Vanessa Ribes.
**Data curation:** Robin Rondon, Vanessa Ribes.

**Formal analysis:** Robin Rondon, Théaud Hezez, Bernadette Drayton-Libotte, Gloria Gonzalez Curto, Claire Dugast-Darzacq, Pascale Gilardi-Hebenstreit, Vanessa Ribes.

**Funding acquisition:** Frédéric Relaix, Vanessa Ribes.

**Investigation:** Robin Rondon, Shinichiro Hayashi, Gloria Gonzalez Curto, Frédéric Auradé, Elie Balloul, Pascale Gilardi-Hebenstreit, Vanessa Ribes.

**Methodology:** Robin Rondon, Théaud Hezez, Julien Richard Albert, Bernadette Drayton-Libotte, Gloria Gonzalez Curto, Frédéric Auradé, Claire Dugast-Darzacq, Pascale Gilardi-Hebenstreit, Vanessa Ribes.

**Project administration:** Frédéric Relaix, Vanessa Ribes.

**Resources:** Frédéric Relaix.

**Supervision:** Claire Dugast-Darzacq, Pascale Gilardi-Hebenstreit, Vanessa Ribes.

**Validation:** Théaud Hezez.

**Visualization:** Robin Rondon, Claire Dugast-Darzacq, Vanessa Ribes.

**Writing – original draft:** Robin Rondon, Pascale Gilardi-Hebenstreit, Vanessa Ribes.

**Writing – review & editing:** Robin Rondon, Théaud Hezez, Shinichiro Hayashi, Claire Dugast-Darzacq, Pascale Gilardi-Hebenstreit.

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
