## [Editor Report · Decision Letter 0]

8 Apr 2025

Dear Dr Ribes,

Thank you for submitting your manuscript entitled "Dual PAX3/7 transcriptional activities spatially encode spinal cell fates through distinct gene networks" for consideration as a Research Article by PLOS Biology and I apologize for the delay in sending you an initial decision. We had wished to discuss your paper with an Academic Editor with relevant expertise, before review, and it took us a bit longer than normal to find someone who was available to provide advice on your study.

Your manuscript has now been evaluated by the PLOS Biology editorial staff as well as by an Academic Editor and I am writing to let you know that we would like to send your submission out for external peer review.

Once your full submission is complete, your paper will undergo a series of checks in preparation for peer review. After your manuscript has passed the checks it will be sent out for review. To provide the metadata for your submission, please Login to Editorial Manager (https://www.editorialmanager.com/pbiology) within two working days, i.e. by Apr 10 2025 11:59PM.

Kind regards,

Luke

Lucas Smith, Ph.D.

Senior Editor

PLOS Biology

lsmith@plos.org

---

## [Decision Letter · Decision Letter 1]

5 Jun 2025

Dear Dr Ribes,

Thank you for your patience while your manuscript "Dual PAX3/7 transcriptional activities spatially encode spinal cell fates through distinct gene networks " was peer-reviewed at PLOS Biology and please accept our apologies for the delay in sending you a decision. Your article was assessed by an academic editor with relevant expertise and we consulted three independent reviewers, although we have, to date, only received reports from two of them. We had been hoping to receive reviewer 3's comments last week, but have still not heard from the last reviewer. However, we feel the two reviewers who have submitted comments already cover the relevant expertise, and after discussion with the Academic Editor, we have decided to forward with a decision based on the feedback we have on hand, in an effort to limit further delays to your manuscript. Please note that we will forward you the third review if it is sent to us belatedly.

In light of the reviews, which you will find at the end of this email, we would like to invite you to revise the work to thoroughly address the reviewers' reports.

As you will see below, the reviewers feel the study represents a potentially important contribution to the field but the have each raised suggestions for how the study could be strengthened further and identify points that need to be clarified. We think that it will be important to thoroughly address the reviewer comments, including by providing additional data and analyses where requested, before we can consider your paper for publication at PLOS Biology.

Given the extent of revision needed, we cannot make a decision about publication until we have seen the revised manuscript and your response to the reviewers' comments. Your revised manuscript is likely to be sent for further evaluation by all or a subset of the reviewers.

**IMPORTANT - SUBMITTING YOUR REVISION**

*Re-submission Checklist*

*Published Peer Review*

*PLOS Data Policy*

*Blot and Gel Data Policy*

Sincerely,

Luke

Lucas Smith, Ph.D.

Senior Editor

PLOS Biology

lsmith@plos.org

REVIEWS:

Reviewer #1: The interplay between extrinsic signalling cues and transcription factor activity in determining cell fate during embryogenesis remains a central question in developmental biology. In this manuscript, Rondon et al. investigate how the transcription factors PAX3 and PAX7, in concert with BMP signalling, contribute to the formation of distinct neural progenitor domains along the dorsal-ventral axis of the neural tube. Employing a combination of mouse genetics, embryonic stem cell-derived organoids, and genomic analyses, the authors demonstrate a crucial synergistic role for PAX3 and PAX7 in ensuring the balanced production of dp1-dp6 interneuron progenitor domains. They further identify downstream transcriptional targets regulated by PAX3/7 and BMP signalling. In addition, the study maps distinct chromatin states associated with these targets and PAX3/7 genomic binding, revealing key enhancer and silencer elements that mediate PAX-driven promotion of dorsal (dp1-3) fates and repression of more ventral identities. Overall, this work represents a significant contribution, supported by well-designed experiments, robust data, and thoughtful interpretation. I have a few comments/questions for the authors:

Major comments:

-The authors propose a compelling model in which BMP signalling in the dorsal neural tube is interpreted through the action of PAX3/7 transcription factors, which recruit enhancer and silencer elements to mediate transcriptional outcomes. While this framework is conceptually appealing, the precise mechanisms by which BMP signals are integrated into this regulatory architecture remain insufficiently defined. Specifically, how BMP activity interacts with PAX transcription factors at the genomic level to specify dorsal cell fates is still an open question. Although the authors discuss potential mechanisms—such as a BMP-induced PAX-ZIC partnership—this aspect of the model would benefit from direct experimental validation. For instance, genome-wide mapping of pSMAD1/5 binding sites in BMP-treated spinal organoids of various genotypes (e.g., PAX3/7 single and double knockouts) could elucidate how BMP signalling interfaces with PAX factors, both individually and in combination. In support of this approach, the observed enrichment of GC-rich motifs in PAX3/7-bound cis-regulatory modules may indicate co-binding with BMP-activated SMAD proteins, consistent with previous findings (Martin-Malpartida et al., 2017; PMID: 29234012).

-In my view, the data presented in Figure 2E do not fully support the authors' claim that "loss of PAX7 had minimal impact on cell composition" (pg. 10). Notably, LHX2 expression appears to be significantly upregulated in PAX7-null organoids compared to BMP4-treated controls, while LBX1 expression is significantly downregulated in PAX7-null organoids specifically in the presence of BMP4. In both cases, the effects of PAX7 loss are opposite to those observed in PAX3-null organoids under the same BMP4 conditions. This raises the possibility of a secondary functional divergence between PAX3 and PAX7 in regulating dorsal neural subtypes. One potential confounding factor is the pooling of data from different clones; variability among individual clones could influence the observed expression patterns. To address this, I recommend that the authors present the data for each clone separately across genotypes. This would clarify whether the observed effects are consistent or driven by clone-specific behaviour. Alternatively, the findings may suggest a previously underappreciated cross-repressive interaction between PAX3 and PAX7, particularly in the specification of dl1 interneuron fate. While the broader model supports cooperative activity between these two transcription factors, the data also point to possible PAX3- and PAX7-specific roles that merit further investigation.

Minor comments:

-in page 8, the authors state that "the balance between the activator and repressor activities of PAX3 and PAX7, regulated along the DV axis, is crucial for shaping the generation of dorsal interneuron subtypes". In my opinion this is the case only for PAX3 as transcriptional activator/repressor versions of PAX7 were not really tested and the statement should be modified accordingly.

-In page 9, the authors claim incorrectly that Pou5f1-negative cells are pluripotent, this should be corrected to Pou5f1+

-In page 9, the authors mention that "DV identities were modulated by BMP4 exposure between days 3-4", however Fig 2 shows that BMP treatment took place between days 3-5. The authors should amend/clarify.

-In Figure 2C. OLIG3+/bgal-negative cells are also shown -how do the authors explain their presence?

-I find Figure S2D confusing: are PAX3+ cells present in PAX3-/- null organoids? What is the explanation for this?

Reviewer #2: Rondon et al present a very substantial original manuscript demonstrating activator (dorsal) and repressor (ventral) functions of PAX3/7 in spinal progenitor patterning using mouse models and organoids. They relate activator functions to BMP responsiveness. This is a very well written manuscript which reports experiments that are both creative and rigorous. A minor concerns should be addressed:

- The statement "My study does not require an ethics statement" is perplexing given the use of mice. This seems adequately covered in the methods section.

- The Pax3 loss of function mice would be expected to develop spina bifida (and excencephaly). This open neural tube phenotype is not apparent in the histology shown, but presumably was seen in later stage embryos or at different anatomical levels? Was the anatomical region analysed predictably outside of the range of this lesion? Did loss of Pax7 modify this phenotype?

- The superimposition of Pax3-GFP on the p34::tk:nLacZ reporter is very elegant. My enthusiasm for it is diminished by the 'overcooked' BGal image in which the background between nuclei is 0. Is reporter activity diminished in Pax3-GFP/GFP embryos or KO organoids?

- Many endpoints in the results are 'normalised' to something, e.g. "normalised to those of heterozygous littermates". The methods used to generate normalised values need to be explained (e.g. for the het normalisation, simple methods such as a ratio versus hets does not appear to have been used).

- Can the authors comment on - or provide data to illustrate - the overlap between pSMAS1/5/8 signal and the regions of the spinal cord in which PAX proteins appear to have gene activation functions? The pSMAD domain in published images of the mouse neural tube appears more dorsally restricted than dp1-3 (e.g. in Escuin et al Dis Models Mech 2023).

- Figure 2E: why are the axes cropped between 60-70% when there are no data points above 70%?

- Figure 2B: The description of BMP4 changing PAX3/7 'expression' seems misleading. The % of cells expressing PAX3 has clearly increased (statistical comparison on the graph would be worthwhile), but the intensity per cell ('expression') appears unchanged. Using different units of measure - cell versus organoid - in the same figure panel is also confusing.

- "Notably, p0-p1… markers were more strongly enriched among PAX-repressed genes, whereas dp4-dp6 lineage markers were predominantly found among PAX-induced genes (Figure 3D)." Where is the p0-p1 data in this figure panel?

- Consider using more figure panel labels in all figures. E.g. Figure 4 has A-E, but could easily be A-G to help the reader understand the different comparisons.

---

## [Decision Letter · Decision Letter 2]

18 Sep 2025

Dear Dr Ribes,

Thank you for your patience while we considered your revised manuscript "Dual PAX3/7 transcriptional activities spatially encode spinal cell fates through distinct gene networks" for publication as a Research Article at PLOS Biology. This revised version of your manuscript has been evaluated by the PLOS Biology editors, the Academic Editor and by one of the original reviewers.

As you will see, reviewer 1 is largely satisfied by the revision, but does have a couple of last points that we think should be addressed before publication. We would therefore like to invite you to address the last reviewer comments, in a revision that we do not think will take very long (but do let us know if you need extra time!). Please also make sure to address the following data and other policy-related requests, detailed below.

**IMPORTANT: In addition to addressing the last comments from Reviewer 1, please address the following editorial requests.

1) TITLE: We think it would be good to spell out PAX3 and PAX7 in the title as that will help people searching for pax7 find the study. We propose you change the title to:

"Dual transcriptional activities of PAX3 and PAX7 spatially encode spinal cell fates through distinct gene networks"

2) DATA: Thank you for providing your sequencing data on GEO. I did not see a reviewer token to access this data (sorry if I missed it somewhere). Can you provide me with one so I can check these data meet our requirements? Please note that this data will need to be set as public by the time of publication.

3) DATA: Thanks also for providing the underlying data for each of your graphs in Supplementary Table S7. Please reference this data in the 'Data Availability Statement' in our editorial manager system. Please also update all relevant figure legends to point readers to this underlying data (you can add the sentence 'the data underlying this figure can be found in Supplementary Table S7').

4) CODE: Per journal policy, if you have generated any custom code during the course of this investigation, please make it available without restrictions. Please ensure that the code is sufficiently well documented and reusable, and that your Data Statement in the Editorial Manager submission system accurately describes where your code can be found.

We expect to receive your revised manuscript within two weeks.

*Published Peer Review History*

*Press*

Sincerely,

Luke

Lucas Smith, Ph.D.

Senior Editor

lsmith@plos.org

PLOS Biology

Reviewer remarks:

Reviewer #1: I am satisfied with the revised version of the manuscript. The authors have added a substantial amount of new data and introduced changes that adequately address my previous comments. I only have two minor suggestions:

1) It would be helpful to include "empty" GFP controls in the data shown in Figure 1G (chick neural tube electroporations).

2_ The sentence "In absence of one paralog, the other one was detected, hence progenitor mainly acquiring a dorsal fate (Figure S3D)" (page 14 of the manuscript) is unclear. I suggest rephrasing for clarity.

---

## [Editor Report · Decision Letter 3]

6 Oct 2025

Dear Vanessa,

Thank you for the submission of your revised Research Article "Dual transcriptional activities of PAX3 and PAX7 spatially encode spinal cell fates through distinct gene networks" for publication in PLOS Biology and thank you for addressing the last reviewer and editorial requests in this revision. On behalf of my colleagues and the Academic Editor, Marianne E. Bronner, I am pleased to say that we can in principle accept your manuscript for publication, provided you address any remaining formatting and reporting issues. These will be detailed in an email you should receive within 2-3 business days from our colleagues in the journal operations team; no action is required from you until then. Please note that we will not be able to formally accept your manuscript and schedule it for publication until you have completed any requested changes.

PRESS

We frequently collaborate with press offices. If your institution or institutions have a press office, please notify them about your upcoming paper at this point, to enable them to help maximize its impact. If the press office is planning to promote your findings, we would be grateful if they could coordinate with biologypress@plos.org. If you have previously opted in to the early version process, we ask that you notify us immediately of any press plans so that we may opt out on your behalf.

Sincerely, 

Luke

Lucas Smith, Ph.D.

Senior Editor

PLOS Biology

lsmith@plos.org